# Divide-and-Conquer Predictive Coding: a Structured Bayesian Inference Algorithm

**Eli Sennesh[1], Hao Wu[2], Tommaso Salvatori[2,3]**
[1]Department of Psychology, Vanderbilt University, Nashville, TN, USA
[2]VERSES AI Research Lab, Los Angeles, USA
[3]Vienna University of Technology, Vienna, Austria
eli.sennesh@vanderbilt.edu, wuhaomxhy@gmail.com, tommaso.salvatori@verses.ai

## Abstract

Unexpected stimuli induce "error" or "surprise" signals in the brain. The theory of predictive coding promises to explain these observations in terms of Bayesian inference by suggesting that the cortex implements variational inference in a probabilistic graphical model. However, when applied to machine learning tasks, this family of algorithms has yet to perform on par with other variational approaches in high-dimensional, structured inference problems. To address this, we introduce a novel predictive coding algorithm for structured generative models, that we call divide-and-conquer predictive coding (DCPC); it differs from other formulations of predictive coding, as it respects the correlation structure of the generative model and provably performs maximum-likelihood updates of model parameters, all without sacrificing biological plausibility. Empirically, DCPC achieves better numerical performance than competing algorithms and provides accurate inference in a number of problems not previously addressed with predictive coding. We provide an open implementation of DCPC in Pyro on Github.

## 1 Introduction

In recent decades, the fields of cognitive science, machine learning, and theoretical neuroscience have borne witness to a flowering of successes in modeling intelligent behavior via statistical learning. Each of these fields has taken a different approach: cognitive science has studied probabilistic *inverse inference* [Chater et al., 2006, Pouget et al., 2013, Lake et al., 2017] in models of each task and environment, machine learning has employed the backpropagation of errors [Rumelhart et al., 1986, Lecun et al., 2015, Schmidhuber, 2015], and neuroscience has hypothesized that *predictive coding* (PC) [Srinivasan et al., 1982, Rao and Ballard, 1999, Friston, 2005, Bastos et al., 2012, Spratling, 2017, Hutchinson and Barrett, 2019, Millidge et al., 2021] may explain neural activity in perceptual tasks. These approaches share in common a commitment to "deep" models, in which task processing emerges from the composition of elementary units.

In machine learning, PC-based algorithms have recently gained popularity for their theoretical potential to provide a more biologically plausible alternative to backpropagation for training neural networks [Salvatori et al., 2023, Song et al., 2024]. However, PC does not perform comparably in these tasks to backpropagation due to limitations in current formulations. First, predictive coding for gradient calculation typically models every node in the computation graph with a Gaussian, and hence fails to express many common generative models. Recent work on PC has addressed this by allowing approximating non-Gaussian energy functions with samples [Pinchetti et al., 2022]. Second, the Laplace approximation to the posterior infers only a maximum-a-posteriori (MAP) estimate and Gaussian covariance for each latent variable, keeping PC from capturing multimodal or correlated distributions. Third, this loose approximation to the posterior distribution results in inaccurate, high-variance updates to the parameters of the generative model.

38th Conference on Neural Information Processing Systems (NeurIPS 2024).

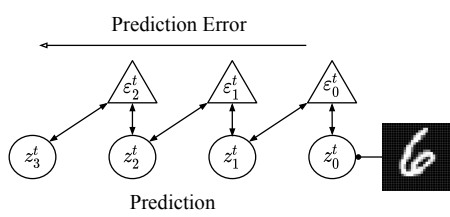
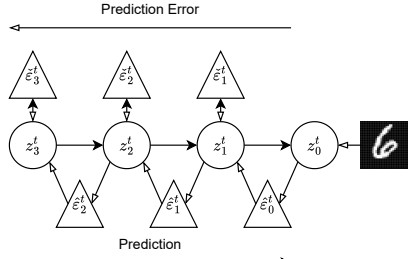

Figure 1: **Left**: Classical PC learns a mean-field approximate posterior with prediction error layers. **Right**: Divide-and-conquer PC approximates the joint posterior with bottom-up and recurrent errors. Where classical predictive coding has layers communicate through shared error units, divide-and-conquer predictive coding separates recurrent from "bottom-up" error pathways to target complete conditional distributions rather than posterior marginal distributions.

In this work we propose a new algorithm, *divide-and-conquer predictive coding* (DCPC), for approximating structured target distributions with populations of Monte Carlo samples. DCPC goes beyond Gaussian assumptions, and decomposes the problem of sampling from structured targets into local coordinate updates to individual random variables. These local updates are informed by unadjusted Langevin proposals parameterized in terms of biologically plausible prediction errors. Nesting the local updates within divide-and-conquer Sequential Monte Carlo [Lindsten et al., 2017, Kuntz et al., 2024] ensures that DCPC can target any statically structured graphical model, while Theorem 2 provides a locally factorized way to learn model parameters by maximum marginal likelihood.

DCPC also provides a computational perspective on the canonical cortical microcircuit [Bastos et al., 2012, 2020, Campagnola et al., 2022] hypothesis in neuroscience. Experiments have suggested that deep laminar layers in the cortical microcircuit represent sensory imagery, while superficial laminar represent raw stimulus information [Bergmann et al., 2024]; experiments in a predictive coding paradigm specifically suggested that the deep layers represent "predictions" while the shallow layers represent "prediction errors". This circuitry could provide the brain with its fast, scalable, generic Bayesian inference capabilities. Figure 1 compares the computational structure of DCPC with that of previous PC models. The following sections detail the contributions of this work:

- Section 3 defines the divide-and-conquer predictive coding algorithm and shows how to use it as a variational inference algorithm;

- Section 4 examines under what assumptions the cortex could plausibly implement DCPC, proving two theorems that contribute to biological plausibility;

- Section 5 demonstrates DCPC experimentally in head-to-head comparisons against recent generative models and inference algorithms from the predictive coding literature.

Section 2 will review the background for Section 3's algorithm: the problem predictive coding aims to solve and a line of recent work adressing that problem from which this paper draws.

## 2   Background

This section reviews the background necessary to construct the divide-and-conquer predictive coding algorithm in Section 3. Let us assume we have a directed, acyclic graphical model with a joint density split into observations $x \in \mathbf{x}$ and latents $z \in \mathbf{z}$, parameterized by some $\theta$ at each conditional density

$$p_\theta(\mathbf{x}, \mathbf{z}) := \prod_{x \in \mathbf{x}} p_\theta(x \mid \mathrm{Pa}(x)) \prod_{z \in \mathbf{z}} p_\theta(z \mid \mathrm{Pa}(z)), \tag{1}$$

where $\mathrm{Pa}(z)$ denotes the parents of the random variable $z \in \mathbf{z}$ and $\mathrm{Ch}(z)$ denotes its children.

**Empirical Bayes**   *Empirical Bayes* consists of jointly estimating, in light of the data, both the parameters $\theta^*$ and the Bayesian posterior over the latent variables $\mathbf{z}$, that is:

$$\theta^* = \arg\max_\theta p_\theta(\mathbf{x}) = \arg\max_\theta \int_{\mathbf{z} \in \mathcal{Z}} p_\theta(\mathbf{x}, \mathbf{z}) \, d\mathbf{z}, \qquad p_{\theta^*}(\mathbf{z} \mid \mathbf{x}) := \frac{p_{\theta^*}(\mathbf{x}, \mathbf{z})}{p_{\theta^*}(\mathbf{x})}.$$

Typically the marginal and posterior densities have no closed form, so learning and inference algorithms treat the joint distribution as a closed-form *un*normalized density over the latent variables; its integral then gives the normalizing constant for approximation

$$\gamma_\theta(\mathbf{z}) := p_\theta(\mathbf{x}, \mathbf{z}), \qquad Z_\theta := \int_{\mathbf{z} \in \mathcal{Z}} \gamma_\theta(\mathbf{z}) \, d\mathbf{z} = p_\theta(\mathbf{x}), \qquad \pi_\theta(\mathbf{z}) := \frac{\gamma_\theta(\mathbf{z})}{Z_\theta}.$$

Neal and Hinton [1998] reduced empirical Bayes to minimization of the *variational free energy*:

$$\mathcal{F}(\theta, q) := \mathbb{E}_{\mathbf{z} \sim q(\mathbf{z})} \left[ -\log \frac{\gamma_\theta(\mathbf{z})}{q(\mathbf{z})} \right] \geq -\log Z(\theta). \tag{2}$$

The ratio of densities in Equation 2 is an example of a *weight* used to approximate a distribution known only up to its normalizing constant. The *proposal* distribution $q(\mathbf{z})$ admits tractable sampling, while the unnormalized *target* density $\gamma_\theta(\mathbf{z})$ admits tractable, pointwise density evaluation.

**Predictive Coding** Computational neuroscientists now often hypothesize that *predictive coding* (PC) can optimize the above family of objective functionals in a local, neuronally plausible way [Millidge et al., 2021, 2023]. More in detail, it is possible to define this class of algorithms as follows:

**Definition 1** (Predictive Coding Algorithm). *Consider approximate inference in a model $p_\theta(\mathbf{x}, \mathbf{z})$ using an algorithm $\mathcal{A}$. Salvatori et al. [2023] calls $\mathcal{A}$ a predictive coding algorithm if and only if:*

1. *It maximizes the model evidence $\log p_\theta(\mathbf{x})$ by minimizing a variational free energy;*

2. *The proposal $q(\mathbf{z}) = \prod_{z \in \mathbf{z}} q(z)$ factorizes via a mean-field approximation; and*

3. *Each proposal factor is a Laplace approximation (i.e. $q_\mu(z) := \mathcal{N}(\mu, \Sigma(\mu))$).*

**Particle Algorithms** In contrast to predictive coding, particle algorithms approach empirical Bayes problems by setting the proposal to a collection of weighted particles $(w^k, \mathbf{z}^k)$ drawn from a sampling algorithm meeting certain conditions (see Definition 4 in Appendix B). Any proposal meeting these conditions (see Proposition 1 in Appendix B and Naesseth et al. [2015], Stites et al. [2021]) defines a free energy functional, analogous to Equation 2 in upper-bounding the model surprisal:

$$\mathcal{F}(\theta, q) := \mathbb{E}_{w, \mathbf{z} \sim q(w, \mathbf{z})} \left[ -\log w \right] \implies \mathcal{F}(\theta, q) \geq -\log Z(\theta).$$

This paper builds on the particle gradient descent (PGD) algorithm of Kuntz et al. [2023], that works as follows: At each iteration $t$, PGD diffuses the particle cloud $q_K(\mathbf{z}) = \frac{1}{K} \sum_{k=1}^K \delta_{\mathbf{z}^k}(\mathbf{z})$ across the target log-density with a learning rate $\eta$ and independent Gaussian noise; it then updates the parameters $\theta$ by ascending the gradient of the log-likelihood, estimated by averaging over the particles. The update rules are then the following:

$$\mathbf{z}^{t+1,k} := \mathbf{z}^{t,k} + \eta \nabla_{\mathbf{z}} \log \gamma_{\theta^t}(\mathbf{z}^{t,k}) + \sqrt{2\eta} \xi^k, \tag{3}$$

$$\theta^{t+1} := \theta^t + \eta \left( \frac{1}{K} \sum_{k=1}^K \nabla_\theta \log \gamma_{\theta^t}(\mathbf{z}^{t+1,k}) \right). \tag{4}$$

The above equations target the joint density of an entire graphical model[1]. When the prior $p_\theta(\mathbf{z})$ factorizes into many separate conditional densities, achieving high inference performance often requires factorizing the inference network or algorithm into conditionals as well [Webb et al., 2018]. Estimating the gradient of the entire log-joint, as in PGD and amortized inference [Dasgupta et al., 2020, Peters et al., 2024], also requires nonlocal backpropagation. To provide a generic inference algorithm for high-dimensional, structured models using only local computations, Section 3 will apply Equation 3 to sample individual random variables in a joint density, combine the coordinate updates via sequential Monte Carlo, and locally estimate gradients for model parameters via Equation 4.

## 3 Divide-and-Conquer Predictive Coding

The previous section provided a mathematical toolbox for constructing Monte Carlo algorithms based on gradient updates and a working definition of predictive coding. This section will combine those

---

[1]Kuntz et al. [2023] also interpreted Equation 3 as an update step along the Wasserstein gradient in the space of probability measures. Appendix C extends this perspective to predictive coding of discrete random variables.

| | PC | LPC | MCPC | DCPC (ours) |
|---|---|---|---|---|
| Generative density | Gaussian | Differentiable | Gaussian | Differentiable |
| Inference approximation | Laplace | Gaussian | Empirical | Empirical |
| Posterior conditional structure | ✗ | ✗ | ✗ | ✓ |

Table 1: Comparison of divide-and-conquer predictive coding (DCPC) against other predictive coding algorithms. DCPC provides the greatest flexibility: arbitrary differentiable generative models, an empirical approximation to the posterior, and sampling according to the target's conditional structure.

tools to generalize the above notion of predictive coding, yielding the novel *divide-and-conquer predictive coding* (DCPC) algorithm. Given a causal graphical model, DCPC will approximate the posterior with a population $q(\mathbf{z})$ of $K$ samples, while also learning $\theta$ explaining the data. This will require deriving local coordinate updates and then parameterizing them in terms of prediction errors.

Let us assume we again have a causal graphical model $p_\theta(\mathbf{x}, \mathbf{z})$ locally parameterized by $\theta$ and factorized (as in Equation 1) into conditional densities for each $x \in \mathbf{x}$ and $z \in \mathbf{z}$. DCPC then requires two hyperparameters: a learning rate $\eta \in \mathbb{R}^+$, and particle count $K \in \mathbb{N}^+$, and is initialized (at $t = 0$) via a population of predictions by ancestor sampling defined as $\mathbf{z}^0 \sim \prod_{z \in \mathbf{z}} p_\theta(z^0 \mid \mathrm{Pa}(z^0))$.

DCPC aims to minimize the variational free energy (Equation 2). The optimal proposal $q_*$ for each random variable would equal, if it had closed form, the *complete conditional* density for that variable, containing all information from other random variables

$$q_*(\mathbf{z}^t \mid \mathbf{z}^{t-1}) \propto \gamma_\theta(z; \mathbf{z}_{\setminus z}) = p_\theta(z \mid \mathrm{Pa}(z)) \prod_{v \in \mathrm{Ch}(z)} p_\theta(v \mid \mathrm{Pa}(v)). \qquad (5)$$

We observe that the prediction errors $\varepsilon_z$ in classical predictive coding, usually defined as the precision weighted difference between predicted and actual value of a variable, can be seen as the *score function* of a Gaussian, where the score is the gradient with respect to the parameter $z$ of the log-likelihood:

$$\varepsilon_z := \nabla_z \log \mathcal{N}(z, \tau) = \tau (x - z);$$

When given the ground-truth parameter $z$, the *expected* score function $\mathbb{E}_{x \sim p(x|z)} [\nabla_z \log p(x \mid z)]$ under the likelihood becomes zero, making score functions a good candidate for implementing predictive coding. We therefore define $\varepsilon_z$ in DCPC as the complete conditional's score function

$$\varepsilon_z := \nabla_z \log \gamma_\theta(z; \mathbf{z}_{\setminus z}) = \nabla_z \log p_\theta(z \mid \mathrm{Pa}(z)) + \sum_{v \in \mathrm{Ch}(z)} \nabla_z \log p_\theta(v \mid \mathrm{Pa}(v)). \qquad (6)$$

This gradient consists of a sum of local prediction-error terms: one for the local "prior" on $z$ and one for each local "likelihood" of a child variable. By defining the prediction error as a sum of local score functions, we write Equation 3 in terms of $\varepsilon_z$ (Equation 6) and the preconditioner of Definition 3:

$$q_\eta(z^t \mid \varepsilon_z^t, z^{t-1}) := \mathcal{N}\left(z^{t-1} + \eta \hat{\Sigma}_\mathcal{I} \varepsilon_z^t, 2\eta \hat{\Sigma}_\mathcal{I}\right).$$

The resulting proposal now targets the complete conditional density (Equation 5), simultaneously meeting the informal requirement of Definition 1 for purely local proposal computations while also "dividing and conquering" the sampling problem into lower-dimensional coordinate updates.

Since the proposal from which we can sample by predictive coding is not the optimal coordinate update, we importance weight for the true complete conditional distribution that is optimal

$$z^t \sim q_\eta(z^t \mid z^{t-1}, \varepsilon_z^t) \qquad\qquad u_z^t = \frac{\gamma_{\theta^{t-1}}(z^t; \mathbf{z}_{\setminus z})}{q_\eta(z^t \mid z^{t-1}, \varepsilon_z^t)}; \qquad (7)$$

resampling with respect to these weights corrects for discretization error, yields particles distributed according to the true complete conditional, and estimates the complete conditional's normalizer

$$\mathrm{RESAMPLE}\left(z^t, u_z^t\right) \sim \pi_{\theta^{t-1}}(z^t \mid \mathbf{z}_{\setminus z}), \qquad\qquad \hat{Z}_{\theta^{t-1}}(\mathbf{z}_{\setminus z})^t := \frac{1}{K} \sum_{k=1}^{K} u_z^{t,k}.$$

The recursive step of "Divide and Conquer" Sequential Monte Carlo [Lindsten et al., 2017, Kuntz et al., 2024] exploits the estimates $\hat{Z}_{\theta^{t-1}}(\mathbf{z}_{\setminus z})^t$ to weigh the samples for the complete target density

$$w_{\theta^{t-1}}^t = \frac{p_{\theta^{t-1}}(\mathbf{x}, \mathbf{z}^t)}{\prod_{z \in \mathbf{z}} \gamma_\theta(z^t; \mathbf{z}_{\setminus z})} \prod_{z \in \mathbf{z}} \hat{Z}_{\theta^{t-1}}(\mathbf{z}_{\setminus z})^t. \qquad (8)$$

---

**Algorithm 1** Divide-and-Conquer Predictive Coding for empirical Bayes

---

**Require:** learning rate $\eta \in \mathbb{R}^+$, particle count $K \in \mathbb{N}$, number of sweeps $S \in \mathbb{N}$
**Require:** initial particle vector $\mathbf{z}^0$, initial parameters $\theta^0$, observations $\mathbf{x} \in \mathcal{X}$

1: **for** $t \in [1 \dots T]$ **do**          ▷ Loop through predictive coding steps
2:   **for** $s \in [1 \dots S]$ **do**       ▷ Loop through Gibbs sweeps over graphical model
3:    **for** $z \in \mathbf{z}$ **do**       ▷ Loop through latent variables in graphical model
4:     $\varepsilon_z \leftarrow \nabla_{\mathbf{z}} \log p_{\theta^{t-1}}(z \mid \mathrm{Pa}(z))$       ▷ Local prediction error
5:     $\varepsilon_z \leftarrow \varepsilon_z + \sum_{v \in \mathrm{Ch}(z)} \nabla_{\mathbf{z}} \log p_{\theta^{t-1}}(v \mid \mathrm{Pa}(v))$   ▷ Children's prediction errors
6:     $\hat{\Sigma}_{\mathcal{I}} \leftarrow \dfrac{\hat{\mathcal{I}}_K(\varepsilon_z^{1:K})^{-1}}{\frac{1}{d}\mathrm{Tr}[\hat{\mathcal{I}}_K(\varepsilon_z^{1:K})^{-1}]}$      ▷ Estimate precision of prediction errors
7:     $z^t \sim q_\eta(z^t \mid \varepsilon_z, z^{t-1})$        ▷ Sample coordinate update
8:     $u_z^t \leftarrow \dfrac{\gamma_{\theta^{t-1}}(z^t; \mathbf{z}_{\setminus z})}{q_\eta(z^t \mid \varepsilon_z, z^{t-1})}$      ▷ Correct coordinate update by weighing
9:     $z^t \leftarrow \mathrm{RESAMPLE}(z^t, u_z^t)$      ▷ Resample from true coordinate update
10:     $\hat{Z}_{\theta^{t-1}}(\mathbf{z}_{\setminus z})^t \leftarrow \frac{1}{K} \sum_{k=1}^{K} u_z^{t,k}$     ▷ Estimate coordinate update's normalizer
11:   $\mathcal{F}^t \leftarrow -\frac{1}{K} \sum_{k=1}^{K} \log \left( \dfrac{p_{\theta^{t-1}}(\mathbf{x}, \mathbf{z}^{t,k})}{\prod_{z \in \mathbf{z}} \gamma_{\theta^{t-1}}(z^{t,k}; \mathbf{z}_{\setminus z}^{t,k})} \prod_{z \in \mathbf{z}} \hat{Z}_{\theta^{t-1}}(\mathbf{z}_{\setminus z})^t \right)$   ▷ Update free energy
12:   $\theta^t \leftarrow \theta^{t-1} + \eta \frac{1}{K} \sum_{k=1}^{K} \nabla_{\theta^{t-1}} \log p_{\theta^{t-1}}(\mathbf{x}, \mathbf{z}^{t,k})$      ▷ Update parameters
13: **return** $\mathbf{z}^T, \theta^T, \mathcal{F}^T$      ▷ Output: updated particles, parameters, free energy

---

By Proposition 1, log-transforming these weights estimates the free energy (Equation 2):

$$\mathcal{F}^t(\mathbf{z}^{t-1}, \theta^{t-1}) := \mathbb{E}_{q*(\mathbf{z}^t \mid \mathbf{z}^{t-1})} \left[ -\log \frac{p_{\theta^{t-1}}(\mathbf{x}, \mathbf{z}^t)}{q_*(\mathbf{z}^t \mid \mathbf{z}^{t-1})} \right] \approx \mathbb{E}_q \left[ -\log w_{\theta^{t-1}}^t \right].$$

Theorem 3 in Appendix B shows that the gradient $\nabla_{\theta^{t-1}} \mathcal{F}^t = \mathbb{E}_q \left[ -\nabla_{\theta^{t-1}} \log p_{\theta^{t-1}}(\mathbf{x}, \mathbf{z}^t) \right]$ of the above estimator equals the expected gradient of the log-joint distribution. Descending this gradient $\theta^t := \theta^{t-1} - \eta \nabla_{\theta^{t-1}} \mathcal{F}^t$ enables DCPC to learn model parameters $\theta$.

The above steps describe a single pass of divide-and-conquer predictive coding over a causal graphical model. Algorithm 1 shows the complete algorithm, consisting of nested iterations over latent variables $z \in \mathbf{z}$ (inner loop) and iterations $t \in T$ (outer loop). DCPC satisfies criteria (1) and (2) of Definition 1, and relaxes criterion (3) to allow gradient-based proposals beyond the Laplace assumption. As with Pinchetti et al. [2022] and Oliviers et al. [2024], relaxing the Laplace assumption enables much greater flexibility in approximating the model's true posterior distribution.

## 4 Biological plausibility

Different works in the literature consider different criteria for biological plausibility. This paper follows the non-spiking predictive coding literature and considers an algorithm biologically plausible if it performs only spatially local computations in a probabilistic graphical model [Whittington and Bogacz, 2017], without requiring a global control of computation. However, while in the standard literature locality is either directly defined in the objective function [Rao and Ballard, 1999], or derived from a mean-field approximation to the joint density [Friston, 2005], showing that the updates of the parameters of DCPC require only local information is not as trivial. To this end, in this section we first formally show that DCPC achieves decentralized inference of latent variables $\mathbf{z}$ (Theorem 1), and then that also the parameters $\theta$ are updated via local information (Theorem 2).

Gibbs sampling provides the most widely-used algorithm for sampling from a high-dimensional probability distribution by local signaling. It consists of successively sampling coordinate updates to individual nodes in the graphical model by targeting their complete conditional densities $\pi_\theta(z \mid \mathbf{x}, \mathbf{z}_{\setminus z})$. Theorem 1 demonstrates that DCPC's coordinate updates approximate Gibbs sampling.

**Theorem 1** (DCPC coordinate updates sample from the true complete conditionals). *Each DCPC coordinate update (Equation 7) for a latent $z \in \mathbf{z}$ samples from $z$'s complete conditional (the normalization of Equation 5). Formally, for every measurable $h : \mathcal{Z} \to \mathbb{R}$, resampled expectations with respect to the DCPC coordinate update equal those with respect to the complete conditional*

$$\mathbb{E}_{z \sim q_\eta(z \mid z^{t-1}, \varepsilon_z^t)} \left[ \mathbb{E}_{u \sim \delta(u), z' \sim \mathrm{RESAMPLE}(z, u_z)} [h(z)] \right] = \int_{z \in \mathcal{Z}} h(z)\, \pi_\theta(z \mid \mathbf{z}_{\setminus z})\, dz.$$

*Proof.* See Corollary 4.1 in Appendix B. □

We follow the canonical cortical microcircuit hypothesis of predictive coding [Bastos et al., 2012, Gillon et al., 2023] or predictive routing [Bastos et al., 2020]. Consider a cortical column representing $z \in \mathbf{z}$; the $\theta$, $\alpha/\beta$, and $\gamma$ frequency bands of neuronal oscillations [Buzsáki and Draguhn, 2004] could synchronize parallelizations (known to exist for simple Gibbs sampling in a causal graphical model [Gonzalez et al., 2011]) of the loops in Algorithm 1. From the innermost to the outermost and following the neurophysiological findings of Bastos et al. [2015], Fries [2015], $\gamma$-band oscillations could synchronize the bottom-up conveyance of prediction errors (lines 4-6) from L2/3 of lower cortical columns to L4 of higher columns, $\beta$-band oscillations could synchronize the top-down conveyance of fresh predictions (implied in passing from $s$ to $s+1$ in the loop of lines 2-9) from L5/6 of higher columns to L1+L6 of lower columns, and $\theta$-band oscillations could synchronize complete attention-directed sampling of stimulus representations (lines 1-11). Figure 5 in Appendix A visualizes these hypotheses for how neuronal areas and connections could implement DCPC.

Biological neurons often spike to represent *changes* in their membrane voltage [Mainen and Sejnowski, 1995, Lundstrom et al., 2008, Forkosh, 2022], and some have even been tested and found to signal the temporal derivative of the logarithm of an underlying signal [Adler and Alon, 2018, Borba et al., 2021]. Theorists have also proposed models [Chavlis and Poirazi, 2021, Moldwin et al., 2021] under which single neurons could calculate gradients internally. In short, if neurons can represent probability densities, as many theoretical proposals and experiments suggest they can, then they can likely also calculate the prediction errors used in DCPC. Theorem 2 will demonstrate that given the "factorization" above, DCPC's model learning requires only local prediction errors.

**Theorem 2** (DCPC parameter learning requires only local gradients in a factorized generative model). *Consider a graphical model factorized according to Equation 1, with the additional assumption that the model parameters $\theta \in \Theta = \prod_{x \in \mathbf{x}} \Theta_x \times \prod_{z \in \mathbf{z}} \Theta_z$ factorize disjointly. Then the gradient $\nabla_\theta \mathcal{F}(\theta, q)$ of DCPC's free energy similarly factorizes into a sum of local particle averages*

$$\nabla_\theta \mathcal{F} = \mathbb{E}_q \left[ -\nabla_\theta \log p_\theta(\mathbf{x}, \mathbf{z}) \right] \approx - \sum_{v \in (\mathbf{x}, \mathbf{z})} \frac{1}{K} \sum_{k=1}^{K} \nabla_{\theta_v} \log p_{\theta_v}(v^k \mid \mathrm{Pa}(v)^k). \tag{9}$$

*Proof.* See Proposition 5 in Appendix B. □

Our practical implementation of DCPC, evaluated in the experiments above, takes advantage of Theorem 2 to save memory by detaching samples from the automatic differentiation graph in the forward ancestor-sampling pass through the generative model.

Finally, DCPC passes from local coordinate updates to the joint target density via an importance resampling operation, requiring that implementations synchronously transmit numerical densities or log-densities for the freshly proposed particle population. While phase-locking to a cortical oscillation may make this biologically feasible, resampling then requires normalizing the weights. Thankfully, divisive normalization appears ubiquitously throughout the brain [Carandini and Heeger, 2012], as well as just the type of "winner-take-all" circuit that implements a softmax function (e.g. for normalizing and resampling importance weights) being ubiquitous in crosstalk between superficial and deep layers of the cortical column [Liu, 1999, Douglas and Martin, 2004].

## 5 Experiments

Divide-and-conquer predictive coding is not the first predictive coding algorithm to incorporate sampling into the inference process, and certainly not the first variational inference algorithm for structured graphical models. This section therefore evaluates DCPC's performance against both models from the predictive coding literature and against a standard deep generative model. Each experiment holds the generative model, dataset, and hyperparameters constant except where noted.

We have implemented DCPC as a variational proposal or "guide" program in the deep probabilistic programming language Pyro [Bingham et al., 2019]; doing so enables us to compute free energy and prediction errors efficiently in graphical models involving neural networks. Since the experiments below involve minibatched subsampling of observations $\mathbf{x} \sim \mathcal{B}$ from a dataset $\mathcal{D} \sim p(\mathcal{D})$ of unknown distribution, we replace Equation 9 with a subsampled form (see Welling and Teh [2011]

| Inference algorithm | Dataset | NLL ↓ | Mean Squared Error ↓ |
|---|---|---|---|
| MCPC | MNIST | $144.6 \pm 0.7$ | $(8.29 \pm 0.05) \times 10^{-2}$ |
| DCPC | MNIST | $\mathbf{102.5 \pm 0.01}$ | $\mathbf{0.01 \pm 7.2 \times 10^{-6}}$ |
| DCPC | EMNIST | $160.8 \pm 0.05$ | $3.3 \times 10^{-6} \pm 3.5 \times 10^{-9}$ |
| DCPC | Fashion MNIST | $284.1 \pm 0.05$ | $0.03 \pm 2.7 \times 10^{-5}$ |

Table 2: Negative log-likelihood and mean squared error for MCPC against DCPC on held-out images from the MNISTs. Means and standard deviations are taken across five random seeds.

for derivation) of the variational Sequential Monte Carlo gradient estimator [Naesseth et al., 2018]

$$\nabla_\theta \mathcal{F} \approx |\mathcal{D}| \mathbb{E}_{\mathcal{B} \sim p(\mathcal{D})} \left[ \frac{1}{|\mathcal{B}|} \sum_{\mathbf{x}^b \in \mathcal{B}} \mathbb{E}_{(\mathbf{z}, w)^{1:K} \sim q} \left[ \log \left( \frac{1}{K} \sum_{k=1}^K w^k \right) \mid \mathbf{x}^b \right] \right]. \tag{10}$$

We optimized the free energy in all experiments using Adam [Kingma and Ba, 2014], making sure to call `detach()` after every Pyro `sample()` operation to implement the purely local gradient calculations of Theorem 2 and Equation 10. The first experiment below considers a hierarchical Gaussian model on three simple datasets. The model consists of two latent codes above an observation.

**Deep latent Gaussian models with predictive coding**   Oliviers et al. [2024] brought together predictive coding with neural sampling hypotheses in a single model: Monte Carlo predictive coding (MCPC). Their inference algorithm functionally backpropagated the score function of a log-likelihood, applying Langevin proposals to sample latent variables from the posterior joint density along the way. They evaluated MCPC's performance on MNIST with a deep latent Gaussian model [Rezende et al., 2014] (DLGM). Their model's conditional densities consisted of nonlinearities followed by linear transformations to parameterize the mean of each Gaussian conditional, with learned covariances. Figure 2 shows that the DLGM structure already requires DCPC to respect hierarchical dependencies.

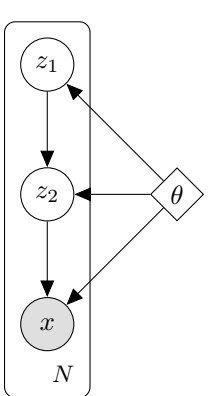

Figure 2: Hierarchical graphical model for DLGM's.

We tested DCPC's performance on elementary reconstruction and generation tasks by using it to train this exact generative model, changing only the likelihood from a discrete Bernoulli to a continuous Bernoulli [Loaiza-Ganem and Cunningham, 2019]. After training we evaluated with a discrete Bernoulli likelihood. Table 2 shows that in terms of both surprise (negative log evidence, with the discrete Bernoulli likelihood) and mean squared reconstruction error, DCPC enjoys better average performance with a lower standard deviation of performance, the latter by an order of magnitude. All experiments used a learning rate $\eta = 0.1$ and $K = 4$ particles.

Figure 3 shows an extension of this experiment to EMNIST [Cohen et al., 2017] and Fashion MNIST [Xiao et al., 2017] as well as the original MNIST, with ground-truth images in the top row and their reconstructions from DCPC-inferred latent codes below. The ground-truth images come from a 10% validation split of each data-set, on which DCPC only infers particles $q_{K=4}(\mathbf{z})$.

The above datasets do not typically challenge a new inference algorithm. The next experiment will thus attempt to learn representations of color images, as in the widely-used variational autoencoder [Kingma and Welling, 2013] framework, without an encoder network or amortized inference.

**Image generation with representation learning**   Zahid et al. [2024] have also recently designed and evaluated Langevin predictive coding (LPC), with differences from both MCPC and DCPC.

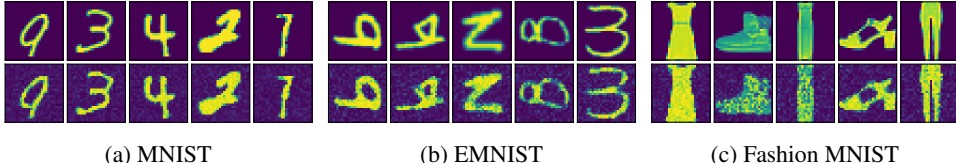

(a) MNIST          (b) EMNIST          (c) Fashion MNIST

Figure 3: **Top**: images from validation sets of MNIST (left), EMNIST (middle), and Fashion MNIST (right). **Bottom**: reconstructions by deep latent Gaussian models trained with DCPC for MNIST (left), EMNIST (middle), and Fashion MNIST (right), averaging over $K = 4$ particles. DCPC achieves quality reconstructions by inference over $\mathbf{z}$ without training an inference network.

| Algorithm | Likelihood | Resolution ↑ | $S\times$ Epochs ↓ | FID ↓ |
|-----------|-----------|--------------|--------------------|-------|
| PGD | $\mathcal{N}$ | $32 \times 32$ | $1 \times 100$ | $100 \pm 2.7$ |
| DCPC (ours) | $\mathcal{N}$ | $32 \times 32$ | $1 \times 100$ | $\mathbf{82.7 \pm 0.9}$ |
| LPC | $\mathcal{DN}$ | $64 \times 64$ | $300 \times 15 = 4500$ | 120 (approximate) |
| VAE | $\mathcal{DN}$ | $64 \times 64$ | $1 \times 4500 = 4500$ | $86.3 \pm 0.03$ |
| DCPC (ours) | $\mathcal{DN}$ | $64 \times 64$ | $30 \times 150 = 4500$ | $\mathbf{79.0 \pm 0.9}$ |

Table 3: FID score comparisons on the CelebA dataset [Liu et al., 2015]. The score for LPC comes from Figure 2 in Zahid et al. [2024], where they ablated warm-starts and initialized from the prior.

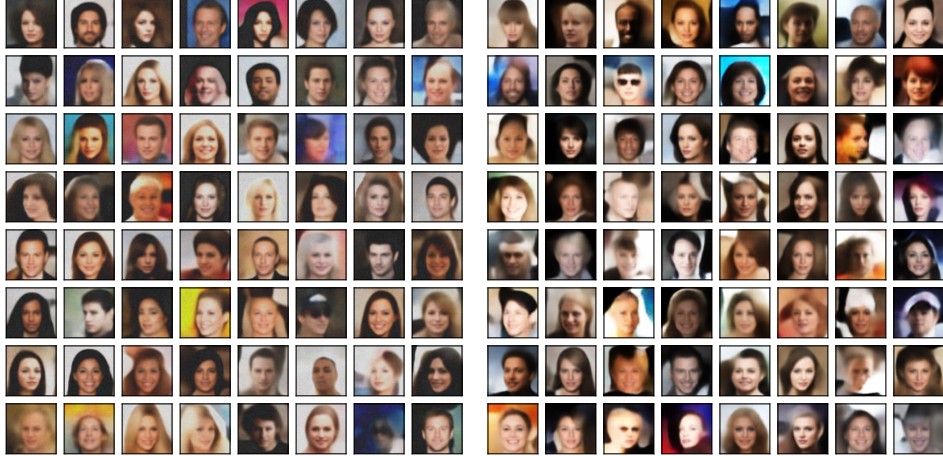

(a) Reconstructions of the CelebA validation set by a generator network trained with DCPC.

(b) Samples drawn *de novo* from the posterior predictive distribution of the trained network.

Figure 4: **Left**: reconstructions from the CelebA validation set. **Right**: samples from the generative model. DCPC achieves quality reconstructions by inference over $\mathbf{z}$ with $K = 16$ particles and no inference network, while the learned generative model captures variation in the data.

While MCPC sends prediction errors up through a hierarchical model, LPC computed as its prediction error the log-joint gradient for all latent variables in the generative model. This meant that biological plausibility, and their goal of amortizing predictive coding inference, restricted them to single-level decoder adapted from Higgins et al. [2017]. We evaluated with their discretized Gaussian likelihood, taken from Cheng et al. [2020], Ho et al. [2020], learning the variance as in Rybkin et al. [2021].

We compare DCPC to LPC using the Frechet Inception Distance (FID) [Seitzer, 2020] featured in Zahid et al. [2024], holding constant the prior, neural network architecture, learning rate on $\theta$, and number of gradient evaluations used to train the parameters $\theta$ and latents $\mathbf{z}$. Zahid et al. [2024] evaluated a variety of scenarios and reported that their training could converge quickly when counted in epochs, but they accumulated gradients of $\theta$ over inference steps. We compare to the results they report after 15 epochs with 300 inference steps applied to latents initialized from the prior, equivalent to $15 \times 300 = 4500$ gradient steps on $\theta$ per batch, replicating their batch size of $64$. Since Algorithm 1 updates $\theta$ only in its outer loop, we set $S = 30$ and ran DCPC for 150 epochs, with $\eta = 0.001$ and $K = 16$. Table 3 shows that DCPC outperforms LPC in apples-to-apples generative quality, though not to the point of matching other model architectures[2] by inference quality alone.

Figure 4 shows reconstructed images from the validation set (left) and samples from the posterior predictive generative model (right). There is blurriness in the reconstructions, as often occurs with variational autoencoders, but DCPC training allows the network to capture background color, hair color, direction in which a face is looking, and other visual properties. Figure 4a shows reconstructions over the validation set, while Figure 4b shows samples from the predictive distribution.

Kuntz et al. [2023] also reported an experiment on CelebA in terms of FID score, at the lower $32 \times 32$ resolution. Since they provided both source code and an exact mathematical description, we were able to run an exact, head-to-head comparison with PGD. The line in Table 3 evaluating DCPC with PGD's example neural architecture at the $32 \times 32$ resolution (with similar particle count and learning rate) demonstrates a significant improvement in FID for DCPC, alongside reduced FID variance.

---

[2]Such as vision Transformers, denoising diffusion models, etc.

**Necessary Compute Resources** The initial DLGM experiments on the MNIST-alike datasets were performed on a desktop workstation with 128GB of RAM and an NVIDIA Quadro P4000 with 8GB of VRAM. Experiments on CelebA were conducted on an NVIDIA DGX equipped with eight (8) NVIDIA A100's, each with 80GB of VRAM. The latter compute infrastructure was also used for unpublished experiments, on several different datasets, in structured time-series modeling.

# 6 Related Work

Pinchetti et al. [2022] expanded predictive coding beyond Gaussian generative models for the first time, applying the resulting algorithm to train variational autoencoders by variational inference and transformer architectures by maximum likelihood. DCPC, in turn, broadens predictive coding to target arbitrary probabilistic graphical models, following the broadening in Salvatori et al. [2022] to arbitrary deterministic computation graphs. DCPC follows on incremental predictive coding [Salvatori et al., 2024] in quickly alternating between updates to random variables and model parameters, giving an incremental EM algorithm [Neal and Hinton, 1998]. Finally, Zahid et al. [2024] and Oliviers et al. [2024] also recognized the analogy between predictive coding's prediction errors and the score functions used in Langevin dynamics for continuous random variables.

There exists a large body of work on how neurobiologically plausible circuits could implement probabilistic inference. Classic work by Shi and Griffiths [2009] provided a biologically plausible implementation of hierarchical inference via importance sampling; DCPC proceeds from importance sampling as a foundation, while parameterizing the proposal distribution via prediction errors. Recent work by Fang et al. [2022] studied neurally plausible algorithms for sampling-based inference with Langevin dynamics, though only for a Gaussian generative model of sparse coding. Golkar et al. [2022] imposed a whitening constraint on a Gaussian generative model for biological plausibility. Finally, Dong and Wu [2023] and Zahid et al. [2024] both suggest mechanisms for employing momentum to reduce gradient noise in a biologically plausible sampling algorithm; the former intriguingly analogize their momentum term to neuronal adaptation. To conclude, other works have already implemented predictive coding models for image generation tasks, a notable example being the neural generative coding framework Ororbia and Kifer [2022], Ororbia and Mali [2022].

# 7 Conclusion

This paper proposed divide-and-conquer predictive coding (DCPC), an algorithm that efficiently and scalably approximates Gibbs samplers by importance sampling; DCPC parameterizes efficient proposals for a model's complete conditional densities using local prediction errors. Section 4 showed how Monte Carlo sampling can implement a form of "prospective configuration" [Song et al., 2024], first inferring a sample from the joint posterior density (Theorem 1) and then updating the generative model without a global backpropagation pass (Theorem 2). Experiments in Section 5 showed that DCPC outperforms the state of the art Monte Carlo Predictive Coding from computational neuroscience, head-to-head, on the simple generative models typically considered in theoretical neuroscience; DCPC also outperforms the particle gradient descent algorithm of Kuntz et al. [2023] while under the constraint of purely local computation. DCPC's Langevin proposals admit the same extension to constrained sample spaces as applied in Hamiltonian Monte Carlo [Brubaker et al., 2012]; our Pyro implementation includes this extension via Pyro's preexisting support for HMC.

DCPC offers a number of ways forward. Particularly, this paper employed naive Langevin proposals, while Dong and Wu [2023], Zahid et al. [2024] applied momentum-based preconditioning to take advantage of the target's geometry. Yin and Ao [2006] demonstrated that gradient flows of this general kind can also provide more efficient samplers by breaking the detailed-balance condition necessary for the Metropolis-Hastings algorithm, motivating the choice of SMC over MCMC to correct proposal bias. Appendix C derives a mathematical background for an extension of DCPC to discrete random variables. Future work could follow Marino et al. [2018], Taniguchi et al. [2022] in using a neural network to iteratively map from particles and prediction errors to proposal parameters.

## 7.1 Limitations

DCPC's main limitations are its longer training time, and greater sensitivity to learning rates, than state-of-the-art amortized variational inference trained end-to-end. Such limitations occur frequently in the literature on neuroscience-inspired learning algorithms, as well as in the literature on particle-based algorithms with no parametric form. This work has no singular ethical concerns specific only to DCPC, rather than the broader implications and responsibilities accompanying advancements in biologically plausible learning and Bayesian inference.

## Acknowledgments and Disclosure of Funding

E.S. was supported by Vanderbilt Brain Institute Faculty Funds, as well as the Vanderbilt Data Science Institute, through PI Andre Bastos. Hamed Nejat produced the laminar circuitry figure in the supplementary material in conjunction with E.S. H.W. and T.S. were supported through VERSES.AI. E.S. would also like to thank Jan-Willem van de Meent, Karen Quigley, and Lisa Feldman Barrett for the training in approximate inference and predictive processing out of which this paper developed.

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

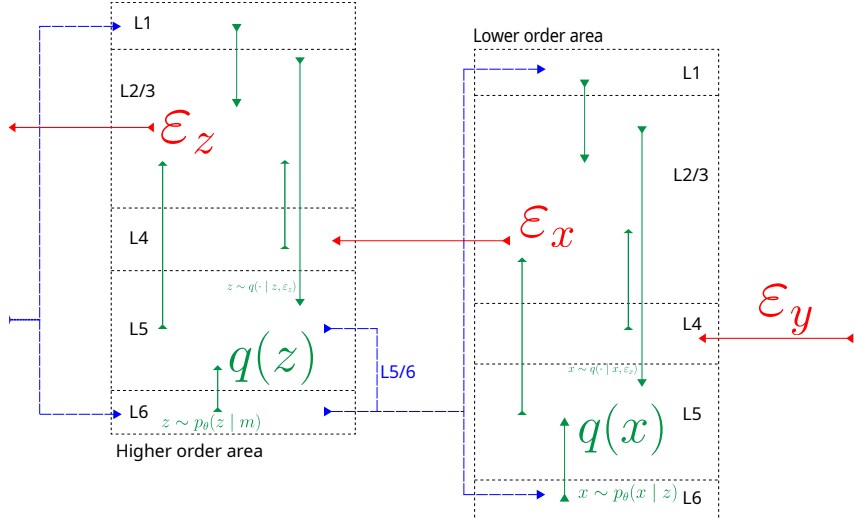

Figure 5: Divide-and-conquer predictive coding provides an algorithmic interpretation for some of the connections mapped in the canonical neocortical microcircuit [Bastos et al., 2012, 2020, Campagnola et al., 2022]: prediction errors (red) arrive through ascending pathways into the central laminar layer 4, which transmits them up to layers 2/3 (green). These layers combine the incoming errors with a present posterior estimate (green L5→ L2/3 connection) to generate prediction errors for the next cortical area. Eventually the updated predictions flow back down the cortical hierarchy (blue).

## A  Further experiments and results

**Alternate image generation/ representation learning**   As indicated in Section 2, this paper builds upon the particle gradient descent (PGD) algorithm; Kuntz et al. [2023] demonstrated the algorithm's performance by training a generator network on CelebA. Their network employed a Gaussian likelihood with a fixed standard deviation of $0.01$, and evaluated a log-joint objective over 100 epochs on exactly 10,000 subsampled data points. The paper then evaluated mean squared error on an inpainting task and the Frechet Inception Distance over data images.

When applied to the resulting target density, DCPC amounts to PGD with a resampling step. Table 4 shows the results of training and evaluating the same model described above with DCPC. Since PGD trained for 100 epochs with a batch size of 128, albeit on a 10,000-image subsample of CelebA, we trained with the entire dataset for 100 epochs with batch-size 128.

| Inference type | Log-joint | FID $\downarrow$ |
|---|---|---|
| VAE ($K = 10$) | $-4.3 \times 10^5$ | $171.5 \pm 0.1$ |
| PGD ($K = 10$) | $-3.8 \times 10^5$ | $100 \pm 2.7$ |
| DCPC (ours, $K = 10$) | $-1.8 \times 10^6$ | $\mathbf{82.7} \pm 0.9$ |

Table 4: Log-joint probabilities and FID metrics show how DCPC performs against the original PGD.

We suspect that the supplied code for log-joint calculation averages over either particles or batch items differently from how we have evaluated DCPC (e.g. we call `mean()` without dividing by any further shape dimensions), accounting for the apparent order-of-magnitude difference between log-joints.

At the request of reviewers, we have substituted a simplified Figure 1 in the main text for Figure 5 showing how to map DCPC onto laminar microcircuit structure.

## B  Importance sampling and gradient estimation proofs

Titsias [2023] introduced optimal estimators for preconditioning Langevin dynamics to adapt with the Fisher information of the target density. Definition 2 gives the most basic estimator for that Fisher information, defined in terms of the score functions we use as prediction errors.

**Definition 2** (Bayesian Fisher estimator [Titsias, 2023]). *Denoting by $\varepsilon_z$ the score function (from Equation 6) and letting $\lambda > 0$ be a fixed hyperparameter, the* Bayesian Fisher estimator

$$\tilde{\mathcal{I}}_K := \mathbb{E}_{\varepsilon_z^{1:K} \sim \pi_\theta(z|\mathbf{z}\setminus z)} \left[ \left( \varepsilon_z \varepsilon_z^\top \right)^k \right] + \frac{\lambda}{K} I \tag{11}$$

*is an empirical estimator of a Bayesian target density's Fisher information (the target density from which the prediction errors were derived) based on a cloud of $K$ particles. When the particles are not yet distributed around a mode of the target (e.g. the score function does not have an average of zero), substituting the empirical covariance for the expected outer product reduces the estimator's bias*

$$\hat{\mathcal{I}}_K(\varepsilon_z^{1:K}) := \mathrm{Cov}\left( \varepsilon_z^{1:K} \right) + \frac{\lambda}{K} I. \tag{12}$$

The above matrix does not describe the preconditioner that Titsias [2023] actually recommended applying in a Langevin proposal. Definition 3 provides the fully normalized preconditioner.

**Definition 3** (Predictive coding Fisher preconditioner [Titsias, 2023]). *Using Definition 2 to parameterize a preconditioner for the Langevin dynamics proposal, the* predictive coding Fisher preconditioner *is the inverse of Equation 12, normalized to have an average eigenvalue of 1*

$$\hat{\Sigma}_{\mathcal{I}}(\varepsilon_z^{1:K}) := \frac{\hat{\mathcal{I}}_K(\varepsilon_z^{1:K})^{-1}}{\frac{1}{d}\mathrm{Tr}[\hat{\mathcal{I}}_K(\varepsilon_z^{1:K})^{-1}]}. \tag{13}$$

Definition 4 generalizes the definition of importance sampling, suitable for recursively constructing sequential Monte Carlo algorithms with changing target densities.

**Definition 4** (Strict proper weighting for a density). *Given an unnormalized density $\gamma_\theta(\mathbf{z})$ with corresponding normalizing constant $Z(\theta)$ and normalized density $\pi_\theta(\mathbf{z})$*

$$Z(\theta) := \int_{\mathbf{z}\in\mathcal{Z}} \gamma_\theta(\mathbf{z})\,d\mathbf{z} \qquad\qquad \pi_\theta(\mathbf{z}) := \frac{\gamma_\theta(\mathbf{z})}{Z(\theta)},$$

*the random variables $w, \mathbf{z} \sim q(w, \mathbf{z})$ are* strictly properly weighted *[Naesseth et al., 2015] with respect to $\gamma_\theta(\mathbf{z})$ if and only if for any measurable test function $h : \mathcal{Z} \to \mathbb{R}$, the weighted expectation over the proposal $q(w, \mathbf{z})$ equals the expectation under the target $\gamma_\theta(\mathbf{z})$*

$$\mathbb{E}_{w,\mathbf{z}\sim q(w,\mathbf{z})}\left[ wh(\mathbf{z}) \right] = \int_{\mathbf{z}\in\mathcal{Z}} h(\mathbf{z})\,\gamma_\theta(\mathbf{z})\,d\mathbf{z}. \tag{14}$$

The following two propositions come from the previous work by Wu et al. [2020], Stites et al. [2021] and Zimmermann et al. [2021]. The reader looking for foundations can see Naesseth et al. [2015] or Chopin and Papaspiliopoulos [2020].

**Proposition 1** (The free energy upper-bounds the surprisal). *Given a proposal $q_\phi(w, \mathbf{z})$ strictly properly weighted (Definition 4) for the target $\gamma_\theta(\mathbf{z})$, the variational free energy provides an upper bound to the target's surprisal*

$$\mathcal{F}(\theta, q) \geq -\log Z(\theta). \tag{15}$$

*Proof.* I begin by writing out the free energy (Equation 2) as an expectation of a negative logarithm

$$\mathcal{F}(\theta, q) = \mathbb{E}_{z,w\sim q(z,w)}\left[ -\log w \right].$$

Jensen's Inequality allows moving the expectation into the negative logarithm by relaxing the definition of the variational free energy from an equality to an upper bound

$$\mathcal{F}(\theta, q) \geq -\log \mathbb{E}_{z,w\sim q(z,w)}\left[ w \right].$$

Setting $h(z) = 1$, strict proper weighting for an unnormalized density (Definition 4) says the expected weight will be the normalizing constant

$$\mathbb{E}_{z,w\sim q(z,w)}\left[ w \right] = Z(\theta)$$

which I substitute back in to obtain the desired inequality $\mathcal{F}(\theta, q) \geq -\log Z(\theta)$. $\qquad\square$

**Proposition 2** (Weighted expectations approximate the normalized target up to a constant). *Given a proposal $q_\phi(w, \mathbf{z})$ strictly properly weighted (Definition 4) for the target $\gamma_\theta(\mathbf{z})$ and a measurable test function $h : \mathcal{Z} \to \mathbb{R}$, weighted expectations under the proposal equal the target's normalizing constant times the test function's expectation under the normalized target*

$$\mathbb{E}_{(w,\mathbf{z}) \sim q_\phi(w,\mathbf{z})} \left[ wh(\mathbf{z}) \right] = Z(\theta) \mathbb{E}_{\mathbf{z} \sim \pi_\theta(\cdot)} \left[ h(\mathbf{z}) \right].$$

*Proof.* Strict proper weighting (Equation 14) states that weighted expectations under the proposal equal integrals over the unnormalized target, and by definition the normalized target equals the unnormalized density over its normalizing constant

$$\mathbb{E}_{w,\mathbf{z} \sim q(w,\mathbf{z})} \left[ wh(\mathbf{z}) \right] = \int_{\mathbf{z} \in \mathcal{Z}} h(\mathbf{z})\, \gamma_\theta(\mathbf{z})\, d\mathbf{z}, \qquad\qquad \pi_\theta(\mathbf{z}) := \frac{\gamma_\theta(\mathbf{z})}{Z(\theta)}.$$

The second equation expresses the unnormalized target in terms of the normalized one

$$Z(\theta)\pi_\theta(\mathbf{z}) = \gamma_\theta(\mathbf{z}),$$

and substituting this expression into the definition of strict proper weighting leads to the desired result

$$\int_{\mathbf{z} \in \mathcal{Z}} h(\mathbf{z})\, \gamma_\theta(\mathbf{z})\, d\mathbf{z} = \int_{\mathbf{z} \in \mathcal{Z}} h(\mathbf{z})\, Z(\theta)\pi_\theta(\mathbf{z})\, d\mathbf{z},$$

$$= Z(\theta) \int_{\mathbf{z} \in \mathcal{Z}} h(\mathbf{z})\, \pi_\theta(\mathbf{z})\, d\mathbf{z}$$

$$\mathbb{E}_{w,\mathbf{z} \sim q(w,\mathbf{z})} \left[ wh(\mathbf{z}) \right] = Z(\theta) \mathbb{E}_{\pi_\theta(\mathbf{z})} \left[ h(\mathbf{z}) \right]. \qquad\qquad \square$$

**Proposition 3** (DCPC's free energy has a pathwise derivative). *The free energy $\mathcal{F}^{t+1} = \mathbb{E}_q \left[ -\log w_{\theta^t}^{t+1} \right]$ constructed by the population predictive coding algorithm (Algorithm 1) has a pathwise derivative as the expectation of the negative gradient of the log-joint density*

$$\nabla_{\theta^t} \mathcal{F}^{t+1} = \mathbb{E}_q \left[ -\nabla_{\theta^t} \log p_{\theta^t}(\mathbf{x}, \mathbf{z}^{t+1}) \right].$$

*Proof.* The free energy has an expression in terms of Equation 8

$$\mathcal{F}^{t+1} = \mathbb{E}_q \left[ -\log w_{\theta^t}^{t+1} \right] \qquad w_{\theta^t}^{t+1} = \frac{p_{\theta^t}(\mathbf{x}, \mathbf{z})}{\prod_{z \in \mathbf{z}} \gamma_\theta(z_b^{t+1}; \mathbf{z}_{\backslash z})} \prod_{z \in \mathbf{z}} \hat{Z}_{\theta^t}(\mathbf{z}_{\backslash z})^{t+1},$$

$$\hat{Z}_{\theta^t}(\mathbf{z}_{\backslash z})^{t+1} = \frac{1}{K} \sum_{k=1}^{K} u_b^{t+1,k} \qquad u_z^{t+1} = \frac{\gamma_\theta(z^{t+1}; \mathbf{z}_{\backslash z})}{q(z^{t+1} \mid \varepsilon_z(z^t))},$$

and writing out the free energy itself in full shows that many terms cancel

$$q(\mathbf{z}^{t+1} \mid \mathbf{z}^t) = \prod_{z_b^{t+1} \in \mathbf{z}^{t+1}} q(z^{t+1} \mid z^t),$$

$$\mathcal{F}^{t+1} = \mathbb{E}_{q(\mathbf{z}^{t+1} \mid \mathbf{z}^t)} \left[ -\log \frac{p_{\theta^t}(\mathbf{x}, \mathbf{z})}{\prod_{z \in \mathbf{z}} \gamma_\theta(z_b^{t+1}; \mathbf{z}_{\backslash z})} \prod_{z \in \mathbf{z}} \frac{1}{K} \sum_{k=1}^{K} \frac{\gamma_\theta(z_b^{t+1}; \mathbf{z}_{\backslash z})}{q(z^{t+1} \mid \varepsilon_z(z^t))} \right]$$

$$= \mathbb{E}_{q(\mathbf{z}^{t+1} \mid \mathbf{z}^t)} \left[ -\log \frac{p_{\theta^t}(\mathbf{x}, \mathbf{z})}{\prod_{z \in \mathbf{z}} \gamma_\theta(z_b^{t+1}; \mathbf{z}_{\backslash z})} \frac{\prod_{z \in \mathbf{z}} \gamma_\theta(z_b^{t+1}; \mathbf{z}_{\backslash z})}{\prod_{z \in \mathbf{z}} q(z^{t+1} \mid \varepsilon_z(z^t))} \right]$$

$$= \mathbb{E}_{q(\mathbf{z}^{t+1} \mid \mathbf{z}^t)} \left[ -\log \frac{p_{\theta^t}(\mathbf{x}, \mathbf{z})}{q(\mathbf{z}^{t+1} \mid \mathbf{z}^t)} \right].$$

The proposal distribution $q$ is a function of the random variable values themselves through the prediction errors, not of the parameters $\theta$. The above expression therefore admits a pathwise

derivative [Schulman et al., 2015], moving the gradient operator into the expectation

$$\nabla_{\theta^t}\mathcal{F}^{t+1} = \nabla_{\theta^t}\mathbb{E}_{q(\mathbf{z}^{t+1}|\mathbf{z}^t)}\left[-\log\frac{p_{\theta^t}(\mathbf{x},\mathbf{z}^{t+1})}{q(\mathbf{z}^{t+1}\mid\mathbf{z}^t)}\right]$$

$$= \mathbb{E}_{q(\mathbf{z}^{t+1}|\mathbf{z}^t)}\left[\nabla_{\theta^t}-\log\frac{p_{\theta^t}(\mathbf{x},\mathbf{z}^{t+1})}{q(\mathbf{z}^{t+1}\mid\mathbf{z}^t)}\right]$$

$$= \mathbb{E}_{q(\mathbf{z}^{t+1}|\mathbf{z}^t)}\left[\nabla_{\theta^t}-\left[\log p_{\theta^t}(\mathbf{x},\mathbf{z}^{t+1})-\log q(\mathbf{z}^{t+1}\mid\mathbf{z}^t)\right]\right]$$

$$= \mathbb{E}_{q(\mathbf{z}^{t+1}|\mathbf{z}^t)}\left[-\left[\nabla_{\theta^t}\log p_{\theta^t}(\mathbf{x},\mathbf{z}^{t+1})-\nabla_{\theta^t}\log q(\mathbf{z}^{t+1}\mid\mathbf{z}^t)\right]\right]$$

$$\nabla_{\theta^t}\mathcal{F}^{t+1} = \mathbb{E}_{q(\mathbf{z}^{t+1}|\mathbf{z}^t)}\left[-\nabla_{\theta^t}\log p_{\theta^t}(\mathbf{x},\mathbf{z}^{t+1})\right]. \qquad\qquad \square$$

**Proposition 4** (DCPC coordinate updates are strictly properly weighted for the complete conditionals).
*Each DCPC coordinate update (Equation 7) for a latent variable $z \in \mathbf{z}$ is strictly properly weighted (Definition 4) for $z$'s unnormalized complete conditional. For every measurable $h : \mathcal{Z} \to \mathbb{R}$*

$$\mathbb{E}_{z\sim q\eta(z^t|z^{t-1},\varepsilon_z^t)}\left[\mathbb{E}_{u\sim\delta(u),z',\hat{Z}\sim\text{RESAMPLE}(z,u_z)}[h(z)]\right] = \int_{z\in\mathcal{Z}} h(z)\,\gamma_\theta(z;\mathbf{z}_{\setminus z})\,dz. \qquad (16)$$

*Proof.* Expanding the outer expectation into an integral and replacing the Dirac delta with the expression for the local weights transforms Equation 16 into

$$\int_{z\in\mathcal{Z}}\frac{\gamma_\theta(z;\mathbf{z}_{\setminus z})}{q\eta(z\mid z^{t-1},\varepsilon_z^t)}\mathbb{E}_{z'\sim\text{RESAMPLE}(z,u_z)}[h(z')]\,q\eta(z\mid z^{t-1},\varepsilon_z^t)\,dz =$$

$$\int_{z\in\mathcal{Z}} h(z)\,\gamma_\theta(z;\mathbf{z}_{\setminus z})\,dz;$$

importance resampling also preserves strict proper weighting (see Naesseth et al. [2015], Stites et al. [2021] and Chopin and Papaspiliopoulos [2020] for proofs), and so this yields

$$\int_{z\in\mathcal{Z}}\mathbb{E}_{z'\sim\text{RESAMPLE}(z,u_z)}[h(z')]\,\gamma_\theta(z;\mathbf{z}_{\setminus z})\,dz = \int_{z\in\mathcal{Z}} h(z)\,\gamma_\theta(z;\mathbf{z}_{\setminus z})\,dz$$

$$\int_{z'\in\mathcal{Z}} h(z')\,\gamma_\theta(z';\mathbf{z}_{\setminus z})\,dz' = \int_{z\in\mathcal{Z}} h(z)\,\gamma_\theta(z;\mathbf{z}_{\setminus z})\,dz.$$

$$\square$$

**Corollary 4.1** (DCPC coordinate updates sample from the true complete conditionals). *Each DCPC coordinate update (Equation 7) for a latent $z \in \mathbf{z}$ samples from $z$'s complete conditional (the normalization of Equation 5). Formally, for every measurable $h : \mathcal{Z} \to \mathbb{R}$, resampled expectations with respect to the DCPC coordinate update equal those with respect to the complete conditional*

$$\mathbb{E}_{z\sim q\eta(z|z^{t-1},\varepsilon_z^t)}\left[\mathbb{E}_{u\sim\delta(u),z'\sim\text{RESAMPLE}(z,u_z)}[h(z')]\right] = \int_{z\in\mathcal{Z}} h(z)\,\pi_\theta(z\mid\mathbf{z}_{\setminus z})\,dz.$$

*Proof.* Proposition 4 in Appendix B provides a lemma

$$\mathbb{E}_{z\sim q\eta(z^t|z^{t-1},\varepsilon_z^t)}\left[\mathbb{E}_{u\sim\delta(u),z',\hat{Z}\sim\text{RESAMPLE}(z,u_z)}[h(z')]\right] = \int_{z\in\mathcal{Z}} h(z)\,\gamma_\theta(z;\mathbf{z}_{\setminus z})\,dz,$$

which we can apply by observing that resampling sums over self-normalized weights

$$\mathbb{E}_{z\sim q\eta(z|z^{t-1},\varepsilon_z^t)}\left[\mathbb{E}_{u\sim\delta(u),z'\sim\text{RESAMPLE}(z,u_z)}[h(z)]\right] =$$

$$\mathbb{E}_{z\sim q\eta(z|z^{t-1},\varepsilon_z^t)}\left[\mathbb{E}_{u\sim\delta(u)}\left[\mathbb{E}_{z'\sim\frac{u\delta_z(\cdot)}{\sum u'}}[h(z')]\right]\right],$$

which is just a weighted sum that by Definition 4 is itself properly weighted

$$\mathbb{E}_{z\sim q\eta(z|z^{t-1},\varepsilon_z^t)}\left[\mathbb{E}_{u\sim\delta(u)}\left[\mathbb{E}_{z'\sim\frac{u\delta_z(\cdot)}{\sum u'}}[h(z')]\right]\right] = \mathbb{E}_{z\sim q\eta(z|z^{t-1},\varepsilon_z^t)}\left[\mathbb{E}_{u\sim\delta(u)}\left[\frac{u}{\sum u}h(z)\right]\right]$$

$$= \mathbb{E}_{z \sim q_\eta(z|z^{t-1}, \varepsilon_z^t)} \left[ \mathbb{E}_{u \sim \delta(u)} \left[ \frac{1}{\sum u} \int_{z \in \mathcal{Z}} h(z) \, \gamma_\theta(z; \mathbf{x}, \mathbf{z}_\backslash) \, dz \right] \right]$$

$$= \mathbb{E}_{z \sim q_\eta(z|z^{t-1}, \varepsilon_z^t)} \left[ \mathbb{E}_{u \sim \delta(u)} \left[ \frac{1}{\cancel{Z_\theta(\mathbf{x}, \mathbf{z}_\backslash)}} \cancel{Z_\theta(\mathbf{x}, \mathbf{z}_\backslash)} \int_{z \in \mathcal{Z}} h(z) \, \pi_\theta(z \mid \mathbf{x}, \mathbf{z}_\backslash) \, dz \right] \right]$$

$$= \int_{z \in \mathcal{Z}} h(z) \, \pi_\theta(z \mid \mathbf{x}, \mathbf{z}_\backslash) \, dz. \qquad \square$$

**Proposition 5** (DCPC parameter learning requires only local gradients in a factorized generative model). *Consider a graphical model factorized according to Equation 1, with the additional assumption that the model parameters $\theta \in \Theta = \prod_{x \in \mathbf{x}} \Theta_x \times \prod_{z \in \mathbf{z}} \Theta_z$ share that factorization. Then the gradient $\nabla_\theta \mathcal{F}(\theta, q)$ of DCPC's free energy similarly factorizes into a sum of local particle averages*

$$\nabla_\theta \mathcal{F} = \mathbb{E}_q \left[ -\nabla_\theta \log p_\theta(\mathbf{x}, \mathbf{z}) \right]$$

$$= \sum_{v \in (\mathbf{x}, \mathbf{z})} \mathbb{E}_{q(v, \mathrm{Pa}(v) | \varepsilon_v, \varepsilon_{\mathrm{Pa}(v)})} \left[ -\nabla_{\theta_v} \log p_{\theta_v}(v \mid \mathrm{Pa}(v)) \right]$$

$$= - \sum_{v \in (\mathbf{x}, \mathbf{z})} \frac{1}{K} \sum_{k=1}^K \nabla_{\theta_v} \log p_{\theta_v}(v^k \mid \mathrm{Pa}(v)^k).$$

*Proof.* Proposition 3 provides the lemma that $\nabla_\theta \mathcal{F} = \mathbb{E}_q \left[ -\nabla_\theta \log p_\theta(\mathbf{x}, \mathbf{z}) \right]$, and applying the factorization of the generative model demonstrates that

$$\nabla_\theta \mathcal{F} = \mathbb{E}_q \left[ -\nabla_\theta \sum_{v \in (\mathbf{x}, \mathbf{z})} \log p_\theta(v \mid \mathrm{Pa}(v)) \right].$$

Since the proposal $q$ does not depend on any $\theta$ and consists of a particle cloud, we can rewrite it as a mixture over the particles (after sampling is performed)

$$\nabla_\theta \mathcal{F} \approx \frac{1}{K} \sum_{k=1}^K -\nabla_\theta \sum_{v \in (\mathbf{x}, \mathbf{z})} \log p_\theta(v^k \mid \mathrm{Pa}(v)^k),$$

and then finally apply the assumption of this theorem that $\theta \in \Theta = \prod_{x \in \mathbf{x}} \Theta_x \times \prod_{z \in \mathbf{z}} \Theta_z$, moving the gradient operation into the sum over individual random variables

$$\approx \frac{1}{K} \sum_{k=1}^K \sum_{v \in (\mathbf{x}, \mathbf{z})} -\nabla_{\theta^v} \log p_{\theta^v}(v^k \mid \mathrm{Pa}(v)^k). \qquad \square$$

## C  Extension to discrete sample spaces

Contemporaneously to the work of Kuntz et al. [2023] on particle gradient descent, Sun et al. [2023] derived a novel Wasserstein gradient flow and corresponding descent algorithm for discrete distributions. In their setting, each Wasserstein gradient step constructs a $D$-dimensional, finitely supported distribution over the $C$-Hamming ball of the starting sample, such that the distribution has $DC$ possible states in total. Let $z^{t+h} \in N_C(z^t)$ denote the resulting discrete random variable in the $C$-neighborhood around $z^t$ with respect to the Hamming distance. The update rule relies on simulating the gradient flow for time $h$, sampling from a Markov jump process at time $t + h$

$$z^{t+h} \sim \prod_{d \in [1 \dots D]} q(z_d^{t+h} \mid z_d^t).$$

A rate matrix $Q_d(z^t)$ defined by the entire discrete variable $z^t$ parameterizes the proposal distribution

$$q_h(z_d^{t+h} \mid z^t) = \exp \left( Q_d(z^t) h \right). \qquad (17)$$

the rate matrix will have nondiagonal entries at indices $i \neq j \in [1 \dots C]$ in the neighborhood $N_C(z^t)$,

$$Q_d(z^t)_{i,j} = w_{i,j} g \left( \frac{\pi_\theta(z_{\backslash d}^t, z_{d,j}')}{\pi_\theta(z_{\backslash d}^t, z_{d,i}')} \right).$$

The above equation requires that $\forall i, j \in [1 \ldots C], w_{i,j} = w_{j,i} \in \mathbb{R}$ and $g(a) = ag\left(\frac{1}{a}\right)$. The ratio of normalized target densities $\pi$ will equal the ratio of unnormalized densities $\gamma$

$$\frac{\pi_\theta(z^t_{\backslash d}, z'_{d,j})}{\pi_\theta(z^t_{\backslash d}, z'_{d,i})} = \frac{\gamma_\theta(z'_{d,j}; z^t_{\backslash d}) \cancel{Z_{z_d}(z^t_{\backslash d}, \theta)}}{\cancel{Z_{z_d}(z^t_{\backslash d})} \gamma_\theta(z'_{d,i}; z^t_{\backslash d})}$$

$$g\left(\frac{\pi_\theta(z^t_{\backslash d}, z'_{d,j})}{\pi_\theta(z^t_{\backslash d}, z'_{d,i})}\right) = g\left(\frac{\gamma_\theta(z'_{d,j}; z^t_{\backslash d})}{\gamma_\theta(z'_{d,i}; z^t_{\backslash d})}\right).$$

Based on the experimental recommendations of Sun et al. [2023], let $w_{i,j} = w_{j,i} = 1$ and $g(a) = \sqrt{a}$. The rate matrix then simplifies to nondiagonal and diagonal terms

$$Q_d(z^t)_{i,j} = \sqrt{\frac{\gamma_\theta(z'_{d,j}; z^t_{\backslash d})}{\gamma_\theta(z'_{d,i}; z^t_{\backslash d})}}, \qquad\qquad Q_d(z^t)_{i,i} = -\sum_{j \neq i} Q_d(z^t)_{i,j}. \qquad (18)$$

Equations 17 and 18 give a distribution descending the Wasserstein gradient of the free energy with respect to a particle cloud in a discrete sample space. Applying Equation 18 to $\gamma_\theta(z; \mathbf{z}_{\backslash z})$ yields a factorization in log space

$$Q(z^t)_{i,j} = \sqrt{\frac{\gamma_\theta(z^t + i; \mathbf{z}^t_{\backslash z})}{\gamma_\theta(z^t + j; \mathbf{z}^t_{\backslash z})}} \quad \log Q(z^t)_{i,j} = \frac{1}{2}\left(\log\gamma_\theta(z^t + i; \mathbf{z}^t_{\backslash z}) - \log\gamma_\theta(z^t + j; \mathbf{z}^t_{\backslash z})\right).$$

This difference can be written as a difference of differences

$$\log\gamma_\theta(z^t + i; \mathbf{z}^t_{\backslash z}) - \log\gamma_\theta(z^t + j; \mathbf{z}^t_{\backslash z}) =$$
$$\left(\log\gamma_\theta(z^t + i; \mathbf{z}^t_{\backslash z}) - \log\gamma_\theta(z^t; \mathbf{z}^t_{\backslash z})\right) - \left(\log\gamma_\theta(z^t + j; \mathbf{z}^t_{\backslash z}) - \log\gamma_\theta(z^t; \mathbf{z}^t_{\backslash z})\right). \quad (19)$$

Recent work on efficient sampling for discrete distributions has focused on approximating density ratios, such as the one in Equation 18, with series expansions parameterized by error vectors. When the underlying discrete densities consist of exponentiating a differentiable energy function, as in Grathwohl et al. [2021], these error vectors have taken the form of gradients and the finite-series expansions have been Taylor series. When they do not, Xiang et al. [2023] showed how they take the form of finite differences and Newton's series

$$\log\gamma(z') - \log\gamma(z) \approx \Delta_1\left(\log\gamma(z)\right)^\top \cdot (z' - z). \qquad (20)$$

Discrete DCPC would therefore use finite differences as discrete prediction errors, breaking each discrete $z \in \mathbf{z}$ into dimensions and incrementing each dimension separately to construct a vector

$$\Delta_1 f(z) := (f(z_1 + 1, z_{2:D}), \ldots, f(z_{1:i}, z_i + 1, z_{i+1:D}), \ldots, f(z_{1:D-1}, z_D + 1)) \ominus f(z), \quad (21)$$

where $\ominus$ subtracts the scalar $f(z)$ from the vector elements and $f : \mathbb{Z}^D \to \mathbb{R}$ is the target function. This would lead to defining the discrete prediction error as the finite difference

$$\varepsilon_z := \Delta_1 \log\gamma_\theta(z^t; \mathbf{z}^t_{\backslash z}). \qquad (22)$$

Applying Equation 20 to the two terms of Equation 19, we obtain the approximations

$$\log\gamma_\theta(z^t + i; \mathbf{z}^t_{\backslash z}) - \log\gamma_\theta(z^t; \mathbf{z}^t_{\backslash z}) \approx \Delta_1\left(\log\gamma_\theta(z^t; \mathbf{z}^t_{\backslash z})\right)^\top \cdot ((z^t + i) - z^t)$$
$$\approx \varepsilon_z(z^t)^\top \cdot i$$

$$\log\gamma_\theta(z^t + j; \mathbf{z}^t_{\backslash z}) - \log\gamma_\theta(z^t; \mathbf{z}^t_{\backslash z}) \approx \Delta_1\left(\log\gamma_\theta(z^t; \mathbf{z}^t_{\backslash z})\right)^\top \cdot ((z^t + j) - z^t)$$
$$\approx \varepsilon_z(z^t)^\top \cdot j,$$

$$\log Q(z^t)_{i,j} \approx \frac{1}{2}\varepsilon_z(z^t)^\top (i - j).$$

Discrete DCPC would thus parameterize its discrete proposal (Equation 17) in terms of $\varepsilon_z$ (Equation 22), so that Equation 18 comes out to the (matrix) exponential of the (elementwise) exponential

$$q_h(z^{t+h} \mid \varepsilon_z) = \exp\left(Q(\varepsilon_z)h\right) \qquad\qquad Q_d(\varepsilon_z)_{i,j} = \exp\left(\frac{(\varepsilon_z)^\top_d (i_d - j_d)}{2}\right).$$

## Supplementary References

Nicolas Chopin and Omiros Papaspiliopoulos. *An Introduction to Sequential Monte Carlo*. Springer, 2020. ISBN 978-3-030-47844-5. doi: 10.1007/978-3-030-47845-2. Citation Key: Chopin2020 ISSN: 2197-568X.

Will Grathwohl, Kevin Swersky, Milad Hashemi, David Duvenaud, and Chris Maddison. Oops I took a gradient: Scalable sampling for discrete distributions. In *Proceedings of the 38th International Conference on Machine Learning*, page 3831–3841. PMLR, July 2021. URL `https://proceedings.mlr.press/v139/grathwohl21a.html`.

John Schulman, Nicolas Heess, Theophane Weber, and Pieter Abbeel. Gradient estimation using stochastic computation graphs. In C. Cortes, N. Lawrence, D. Lee, M. Sugiyama, and R. Garnett, editors, *Advances in Neural Information Processing Systems*, volume 28. Curran Associates, Inc., 2015. URL `https://proceedings.neurips.cc/paper_files/paper/2015/file/de03beffeed9da5f3639a621bcab5dd4-Paper.pdf`.

Haoran Sun, Hanjun Dai, Bo Dai, Haomin Zhou, and Dale Schuurmans. Discrete Langevin Samplers via Wasserstein Gradient Flow. In *Proceedings of the 26th International Conference on Artificial Intelligence and Statistics*, Valencia, Spain, April 2023. Proceedings of Machine Learning Research.

Michalis Titsias. Optimal preconditioning and fisher adaptive langevin sampling. In *Advances in Neural Information Processing Systems*, volume 37. Curran Associates, Inc., 2023. URL `https://proceedings.neurips.cc/paper_files/paper/2023/hash/5da6d5818a156791090c875abeca3cf8-Abstract-Conference.html`.

Hao Wu, Heiko Zimmermann, Eli Sennesh, Tuan Anh Le, and Jan Willem van de Meent. Amortized population Gibbs samplers with neural sufficient statistics. In *Proceedings of the 37th International Conference on Machine Learning*, 2020.

Yue Xiang, Dongyao Zhu, Bowen Lei, Dongkuan Xu, and Ruqi Zhang. Efficient Informed Proposals for Discrete Distributions via Newton's Series Approximation. In *Proceedings of The 26th International Conference on Artificial Intelligence and Statistics*, pages 7288–7310. PMLR, April 2023. URL `https://proceedings.mlr.press/v206/xiang23a.html`. ISSN: 2640-3498.

Heiko Zimmermann, Hao Wu, Babak Esmaeili, and Jan-Willem van de Meent. Nested variational inference. In M. Ranzato, A. Beygelzimer, Y. Dauphin, P.S. Liang, and J. Wortman Vaughan, editors, *Advances in Neural Information Processing Systems*, volume 34, pages 20423–20435. Curran Associates, Inc., 2021. URL `https://proceedings.neurips.cc/paper_files/paper/2021/file/ab49b208848abe14418090d95df0d590-Paper.pdf`.

