# OpenReview forum: "Divide-and-Conquer Predictive Coding: a structured Bayesian inference algorithm"
_NeurIPS.cc/2024/Conference — NeurIPS 2024 poster_

### Official Review · Reviewer_Cx3n · 2024-06-19

**Soundness:** 3
**Presentation:** 3
**Contribution:** 3
**Rating:** 7
**Confidence:** 2

**Summary:**

This paper proposes a version of predictive coding that is biologically plausible (in the sense that all computations are local), and can perform both classification and generation (by operating in both directions). This version of predictive coding, dubbed Population Predictive Coding (PPC), overcomes the limitations of traditional PC by enabling a more general class of distributions in the latent layers.

**Strengths:**

The work seems to have a rigorous probabilistic coverage of predictive coding.

**Weaknesses:**

Some notation needs to be defined to help the reader. For example:
- eqn (1): the function Pa is not defined.
- eqn (5): Ch(z) is not defined. I infer from Algorithm 1 that it probably means "children of", but it should be made clear when it is first used.


Additionally, there are a few things that could be fixed.
- There is no figure 3.
- figure 4 has 4 rows, but the caption only states what is in the "Top" and "Bottom". Please clarify.

**Questions:**

I would have liked to see the classification accuracy of the model on MNIST (as a percent). Predictive coding networks tend not to perform well on classification tasks.

**Limitations:**

Yes

---

> ### Author Rebuttal · Authors · 2024-08-07
>
> We thank the reviewer for the time and valuable feedback.
>
> > would have liked to see the classification accuracy of the model on MNIST (as a percent). Predictive coding networks tend not to perform well on classification tasks.
>
> It is true that, when used as generative models, predictive coding networks do not perform well on image classification tasks. However, when used in a discriminative direction (where the goal is to generate the label, providing the image as a very precise (Delta) prior on the first layer), PC has been shown to perform as well as backprop in complex image classification tasks:
>
> https://arxiv.org/abs/2407.01163
>
> However, the study of discriminative PCNs is not the focus of our work, and we have hence not considered it in our tests.
>
> > Typos
>
> Thank you for the pointers, they have now been addressed in the final version of the manuscript.

---

> > ### Comment · Reviewer_Cx3n · 2024-08-07
> > **Thank you**
> >
> > I have read your responses.
> >
> > In the paper cited above, the PC networks achieved high classification accuracy (approx 98%), but their architectures had the predictive connections projection from the image inputs toward the classification layer, similar to [Whittington, Bogacz; 2017]. Is that the same as the models in this paper? Which directions do the predictions flow? From input toward latent (classification vector), or from latent toward the input (images)?

---

> > > ### Author Response · Authors · 2024-08-07
> > > **This paper studies generative modeling.**
> > >
> > > Thank you for your quick response.
> > >
> > > This paper studies generative modeling, in which predictions flow top-down from a latent (initialized from the prior and then updated towards the posterior by predictive coding) toward the data (images).
> > >
> > > In greater detail, this paper studied predictive coding as an inference strategy in Bayesian generative modeling, not PCNs as a neural network architecture.  That is why we have not conducted a classification experiment.

---

### Official Review · Reviewer_DHBv · 2024-07-10

**Soundness:** 3
**Presentation:** 3
**Contribution:** 3
**Rating:** 6
**Confidence:** 3

**Summary:**

SETTING: *Biologically plausible* EM with variational inference in directed graphical models.  In particular, the authors aim to implement predictive coding with local parameter updates for learning.


APPROACH: "Variational" EM with sampling (sequential Monte Carlo, SMC) in place of a separate inference/recognition model---a method called "particle gradient descent" (PGD, Kuntz et al., AISTATS 2023).

More precisely, following Neal & Hinton, the authors formulate the objective as minimizing the free energy (i.e. maximizing the ELBo), an upper bound on the marginal cross entropy (the fit of the model to the data).  In place of the recognition model (as in Neal & Hinton, VAEs, etc.), PGD implements Langevin dynamics (LD) in latent space.  More precisely still, a single step of LD is interpreted as providing a (Gaussian) proposal density, for use in importance sampling, with the unnormalized posterior of the generative model as the target distribution.  The importance weights are then used to resample the particles (making this into an SMC algorithm).

To handle structured probabilistic graphical models, this entire procedure is embedded into a Gibbs sampler.  That is, the particle representation for a particular latent variable is computed with all other latent variables fixed; the sampling procedure is repeated for each latent variable in turn; and *this* entire procedure, which constitutes a single step of Gibbs sampling, is repeated S times.  Then the free energy and model parameters are updated once.  *This* entire update procedure is itself repeated T times, with the LD for each particle continuing from where it left off.

The MS aims to produce a biologically plausible algorithm, and identifies its steps with biologically plausible operations:

--The Gibbs sampling is proposed to be coordinated with cortical oscillations.

--Each cortical column is proposed to encode an individual latent variable, with columnar connectivity reflecting the structure of the directed graph.  Thus columns receive their parents' current (sampled) state via L1 or L6 in order to compute conditional probabilities; locally calculate the energy gradients, and pass it "up" (L2/3 ->) to their parents (-> L4) for accumulation.

--Parameter updates can all be performed locally.

--Computation of the joint from the local complete conditionals is coordinated by phase-locking to cortical oscillations (at some other frequency?).

**Strengths:**

The algorithm provides an interesting approach to spatially localizing computations, a prerequisite for mapping to (real) neural circtuits.  Models learned with the algorithm can reconstruct images from simple data sets.

**Weaknesses:**

This MS has two major weaknesses:

(1) The results are very thin/missing.  Indeed, the only results reported in the whole MS are reconstructions (Figs. 2 and 5, Table 2), and these are quantified only for MNIST (and cousins).  They are certainly not close to the performance of modern generative models.

Several additional results or their descriptions are missing:

--Table 3 is missing the FID score!

--There is no Figure 3.

--Figure 4 is not referenced in the text.

--Figure 5 refers to "top" and "bottom," but there are four rows of images.  (And it is not clear visually which images are reconstructions of which other images.)

I also expected to see an example with a graphical model that has some actual structure, since this is part of the appeal of the algorithm, but there is none.

(2) It is difficult to determine precisely what aspects of the method are new and not from published work (especially Kuntz 2023, Kuntz 2024, Naesseth 2015).  If the algorithm exists in prior work and the contribution of this MS is supposed to be the mapping to biological circuits, then it does not provide much novelty at all (more on this in the questions below).

**Questions:**

--It is difficult to determine precisely what aspects of the method are new and not from published work (especially Kuntz 2023, Kuntz 2024, Naesseth 2015).  Can the authors clarify this?  (I have only read these papers cursorily.)

--Is the mapping to the cortical microcircuit and other neurological constraints supposed to be the (or a) main result of the MS?  These are certainly intriguing but they are not very constraining.  Can the authors make predictions for neuroscience experimentalists based on their mappings?  Can they explain empirical findings in the literature?

--Is it necessary to write the free energy in terms of weights?  Why not just let the recognition model be the product over the normalized "complete conditionals" (with the normalizers computed with Langevin dynamics)?

--The authors propose cortical oscillations as the mechanism for synchronizing a parallel form of Gibbs sampling (that is still guaranteed to have the right stationary distribution).  At what frequency do they hypothesize these oscillations to be?  Given a reasonable number of Gibbs sweeps, does this let the computation happen fast enough?

More generally, can the authors give some more detail on the time complexity of the algorithm and how they would expect it to scale to problems with (say) real graph structure?

--Proposition 5 is repeated as Corollary 5.1, which then references Proposition 5 (should be 3).

--"PPC" is already the name of a (very popular) proposed representation of probability distributions in populations of neurons ("probabilistic population codes," Ma et al., Nature Neuroscience, 2006).  Perhaps the authors can invent another name.

---

> ### Author Rebuttal · Authors · 2024-08-07
>
> We thank the reviewer for the time and valuable feedback.
>
> > Novelty over previous work (Kuntz 2023;2024, Naesseth 2015)
>
> Kuntz 2023 did not construct or evaluate importance weights, while Lindsten 2017/Kuntz 2024 and Naesseth 2015 did not propose a unique decomposition of a target model into Gibbs kernels as target densities for samplers, nor did they employ gradient-based proposals. Our DCPC algorithm starts from a neuroscientific motivation to derive a new sampling algorithm: first approximate the true Gibbs kernels using gradient-based proposals and SMC, then take the nested D&C-SMC step “up” to calculate importance weights for the entire joint density, then take the pathwise derivative of their negative logarithm to estimate gradients of the complete generative model’s free-energy. Finally, as far as we know, ours is the first sampling-based predictive coding algorithm to compute a free energy that properly upper-bounds the surprisal of the sensory data, as required by the Free Energy Principle and achieved in neuronal message-passing proposals.
>
> > Is the mapping to the cortical microcircuit and other neurological constraints supposed to be the (or a) main result of the MS? These are certainly intriguing but they are not very constraining. Can the authors make predictions for neuroscience experimentalists based on their mappings? Can they explain empirical findings in the literature?
>
> The mapping to the cortical microcircuit is not intended to be the main result of the manuscript, only a suggestion. The primary contribution is the algorithm itself, which should stand or fall on its own.
>
> > The authors propose cortical oscillations as the mechanism for synchronizing a parallel form of Gibbs sampling. At what frequency do they hypothesize these oscillations to be? Given a reasonable number of Gibbs sweeps, does this let the computation happen fast enough?
>
> The γ-band of cortical oscillations take place at 30–150 Hz and tend to correlate with bottom-up processing of oddball or deviant stimuli, in contrast to the top-down alpha/beta band in the 8–30 Hz range.  “High gamma” past 50 Hz is often more associated with “prediction error”. Without having the experimental evidence to break down gamma more finely, the oscillation frequency band we can associate with “prediction error” is faster than the frequency band associated with “prediction”. Picking the median of each frequency band, median alpha/beta would be 19 Hz while median gamma would be 90 Hz, more than 4x faster. We do suggest that if inference takes place 4x faster than generative posterior prediction, with a reasonably efficient inference algorithm, inference can converge fast enough.
>
> > More generally, can the authors give some more detail on the time complexity of the algorithm and how they would expect it to scale to problems with (say) real graph structure?
>
> Abstracting over the time complexity of automatic differentiation and SMC, our algorithm has linear asymptotic complexity in the number of nodes in a graph, and then multiplicative factors in the number of sweeps and the total number of inference steps in the training loop. Further asymptotic guarantees would require details of model structure (ie: log-concavity, etc).
>
> > The results are very thin/missing.
>
> We have now updated the manuscript, and added multiple, generative experiments on the Celeb64 dataset. Note that, while the results are not as good as the ones of modern generative models, they perform as well (and sometimes outperform) these of bio-plausible learning algorithms, such as other predictive coding networks such as Oliver 2024: https://www.biorxiv.org/content/10.1101/2024.02.29.581455v1.full
> and the particle gradient descent antecedent to ours. For more details about the experiments, we refer to the paragraph Numerical and Figure Results on CelebA, provided as a general answer to all the reviewers.
>
> > Is it necessary to write the free energy in terms of weights? Why not just let the recognition model be the product over the normalized "complete conditionals" (with the normalizers computed with Langevin dynamics)?
>
> It is indeed mathematically necessary to write the free energy in terms of importance weights targeting the generative joint density. By definition, a variational free energy is the expected value of the negative logarithm of an importance weight.  Equivalently, the VFE is the cross-entropy of the generative joint distribution, taken with respect to the proposal distribution (recognition model), minus the entropy of the recognition model.  The product of normalized complete conditionals (with the normalizers estimated by importance sampling) would only estimate the entropy of the recognition model, the second term.
>
> > Typos
>
> Thank you for the pointers, they have now been addressed in the final version of the manuscript.

---

> > ### Comment · Reviewer_DHBv · 2024-08-12
> >
> > The authors have filled in the missing results, and distinguished their contribution from other papers, so I have raised my score.
> >
> > > It is indeed mathematically necessary to write the free energy in terms of importance weights targeting the generative joint density. By definition, a variational free energy is the expected value of the negative logarithm of an importance weight. Equivalently, the VFE is the cross-entropy of the generative joint distribution, taken with respect to the proposal distribution (recognition model), minus the entropy of the recognition model. The product of normalized complete conditionals (with the normalizers estimated by importance sampling) would only estimate the entropy of the recognition model, the second term.
> >
> > I was not proposing to drop either term from the free energy, but rather to rewrite it without any reference to "weights" or "strictly properly weighting."  That is, precisely to rewrite it as "the cross-entropy of the generative joint distribution, taken with respect to the proposal distribution (recognition model), minus the entropy of the recognition model," where the recognition model is defined to be the product over the normalized "complete conditionals" (with the normalizers computed with Langevin dynamics).  This would (in this reviewer's opinion) substantially simplify the presentation.  But perhaps there is some obstacle to writing it this way that I am overlooking.

---

> > > ### Author Response · Authors · 2024-08-12
> > >
> > > Aha, we seem to have misunderstood what you were saying.  At least this author had always thought of a free-energy as definitionally involving a proposal/recognition model q that admitted closed-form sampling, providing a base-case on which the more elaborate sampling algorithms are written.  From the point of view of how you're putting it, yes, we could write the free energy in terms of a product of complete conditionals as the recognition model Q and then consider *estimating* it by Monte Carlo with the whole Langevin to nested SMC procedure.  We agree with the reviewer that from the point of view of optimization rather than sampling algorithms, that would likely be much clearer, and will adopt that into the manuscript.

---

> > > > ### Comment · Reviewer_DHBv · 2024-08-13
> > > >
> > > > Ok, good, we're on the same page now.  To me this formulation is simpler, but I defer to the authors' judgment here.

---

### Official Review · Reviewer_AbQA · 2024-07-12

**Soundness:** 3
**Presentation:** 3
**Contribution:** 3
**Rating:** 7
**Confidence:** 4

**Summary:**

Predictive coding (PC) is a speculative but attractive theory of how certain parts of the brain—especially the so-called 'canonical' cortical microcircuit—might implement Bayesian inference, and hence focus processing effort on the 'surprising' rather than the 'predictable' features of (e.g., sensory) stimuli. PC requires a generative model and an optimization algorithm (i.e., given observations, what is the posterior associated with the model's parameters?).

While there exists a rich literature related to how PC might be implemented in the brain, previous work has been biologically implausible in a few different senses. In particular, usually (i) a simple (usually Gaussian) generative model is assumed; and/or (ii) inference involves some kind of approximation; and/or (iii) the algorithm is not necessarily fully 'local'.

The authors propose a novel implementation of PC that they argue is biologically plausible. They call it "population prediction coding" (PPC). They also show that their algorithm can solve some practical machine learning tasks.

**Strengths:**

The authors study an interesting topic and display a good grasp of the literature. There are a lot of interesting ideas referenced and discussed throughout.

**Weaknesses:**

My main concerns are that the core ideas of the paper are confusingly presented, that the 'biological plausibility' claim may be too strong, and that the model's performance is a bit underwhelming.

A more minor concern is that figure real-estate is used strangely (why do 80 copies of the same picture take up a huge amount of space in Fig. 4?). It's 'cool', but why is the cortical microcircuit figure (Figure 1) there? It doesn't seem like any of its details are referenced in the main text. Much more helpful would be a diagram of the proposed PPC algorithm, or better yet, two side-by-side diagrams comparing the proposed PPC algorithm to a more 'vanilla' PC algorithm.

A very minor quibble: "PPC" in neuroscience is already used to refer to "probabilistic population codes" (see Ma et al. 2006) as well as the "posterior parietal cortex", so using PPC here is a bit confusing.

Another relatively minor point. Does vanilla PC really *require* Gaussian densities and the Laplace approximation (Table 1)? My understanding is that the framework itself doesn't *require* this; rather, this is just a useful assumption people use to get something simple and workable. I could be wrong, though. Discussing this point in the paper would be very helpful.

**Confusing presentation.** The meat of the work is in Sections 2 through 4, and I found these sections confusingly written. In Sections 2 and 3, this is probably in part because the authors have to both (a) discuss standard PC in detail, and (b) introduce their modifications to the standard idea. The way things were written made it hard for me to tell what's 'new' and what's not, so it would be extremely helpful if the sections of the text were very obviously separated between 'this is old PC' and 'this is a new modification to PC we are proposing'. As mentioned above, a diagram of vanilla PC vs the proposed modification to it would greatly help.

In a few places, the motivation for doing things a certain way is also not quite clear. For example, PPC involves using particle-based gradient descent; why? Why this particular approach to empirical Bayes and not some other one? Are there good reasons to believe this to be a good basis for a 'local' solution to empirical Bayes? This does not appear to be discussed in the text. Put differently, is it better to think of this as a *possible* way the brain could implement empirical Bayes, or for some reason a *really good candidate* for that?

**Biological plausibility.** This term is fraught and my view here is one that others may not agree with. But I think the authors are overclaiming here when they discuss biological plausibility. While they acknowledge that it means different things to different people (line 145), I still think this is insufficient.

The authors really mean *local computations* when they talk about biological plausibility, as they make clear between lines 145 and line 152. But biologically plausible neural computations involve neurons, and hence I think it is reasonable to expect that there is some discussion of whether or not neurons can implement the computations required. Locality is a start, but it is not the end, and in general it can be quite a bit of work to convert the steps of an algorithm into something neurons can plausibly do. There are references suggesting that neurons *might* be able to learn the required log-density gradients (line 175), but this is more of an afterthought than a core part of the contribution of this work, and these thoughts are not well-developed. Without a more specific proposal for how neurons can learn log-density gradients, and some in-silico validation that the proposed strategy works, I really think the term "biological plausibility" should be removed in many places, or else used with extreme caveats included near each use.

Finally, part of the goal of developing 'biologically plausible' algorithms is to make predictions about biology. What are the specific predictions here? If there are some interesting ones, how do they compare to the predictions of other variants of PC? This should be explicitly discussed.

**Underwhelming performance.** First, I found the tasks used in Sec. 5 confusingly described. It would be helpful if the details of the reconstruction and generation tasks, as well as the training procedure, were better explained in the main text. Second, there are not that many comparisons shown in Table 2; are there other alternatives PPC could be compared to? Third, in Table 3, PPC's FID score is shown as a question mark; I am not sure if this is an error. I assume "undefined" would not be good.

Finally, there are some issues related to training time and speed. Can plots analyzing these features be included? How do other PC approaches compare on these axes? (Is there some time - performance tradeoff, for example?)

**Questions:**

1. Can the authors briefly summarize the 'new' features of their algorithm relative to other PC ideas?
2. Why particle-based gradient descent as opposed to some other approach to empirical Bayes?
3. Concretely, how might neurons learn log-density gradients?
4. What is the FID score of PPC in Table 3?
5. Is PPC strictly better than other PC algorithms, or is there some kind of tradeoff?

**Limitations:**

The authors discuss some limitations (line 296). I think they could have better discussed limitations related to whether neurons can perform the desired computations, and whether particle-based gradient descent is the only / most appropriate possibility here. I think possible tradeoffs related to (e.g., training) time could also be interesting to discuss.

---

> ### Author Rebuttal · Authors · 2024-08-07
>
> We thank the reviewer for the time and valuable feedback.
>
> > Does vanilla PC really require Gaussian densities and the Laplace approximation (Table 1)? My understanding is that the framework itself doesn't require this; rather, this is just a useful assumption people use to get something simple and workable. I could be wrong, though.
>
> We searched the literature for a definition that would clarify whether vanilla PC requires or merely uses the Laplace approximation and Gaussian densities. The definition we found in Salvatori et al 2023, now included and discussed in the manuscript, says that they are required. On a somewhat more in-depth level, Bogacz 2017 and a number of papers by Friston appear to imply that the Free Energy Principle literature first constructs the VFE/ELBO as is typically done in machine learning, and then applies the Laplace approximation and Jensen’s Inequality to the VFE itself.  The resulting energy function provides a looser bound on the model evidence than the usual ELBO in machine learning, but (seemingly vitally to this earlier work) saves the resulting PC schemes from having to perform any Monte Carlo computations. Our use of up-to-date SMC and variational inference methods to construct our VFE/ELBO and evaluate its gradients appear a required ingredient for relaxing PC’s Laplace assumption.
>
> > The authors really mean local computations when they talk about biological plausibility … But biologically plausible neural computations involve neurons, and hence I think it is reasonable to expect that there is some discussion of whether or not neurons can implement the computations required.
>
> We agree that the term “bio-plausibility” has been loosely adopted in the literature, often indeed to refer to purely local computations. In our case, we consider our list of minimal properties to consist of local computations and the lack of a global control or synchronization signal. As much of the field refers to predictive coding models as biologically plausible, we follow the mainstream view and keep this terminology for our variation upon PC. We have also added a discussion about biological plausibility in the manuscript that specifies where we are referring to local computations, as well as references to a number of papers showing that neurons across a variety of model organisms (up to and including primate cortex) can calculate derivatives of logarithms of underlying signals. We thank the reviewer for the opportunity to go into detail on an underappreciated question in the study of predictive coding algorithms.
>
> > There are some issues related to training time and speed. Can plots analyzing these features be included?
>
> Yes. Given some time we can retrieve that data from the training logs and plot it. At an approximate estimate, since PPC has to load and unload per-batch particles from GPU memory at every iteration, it takes time to train that is more on par with minibatched mean-field variational approaches than amortized variational approaches, approximately 4x the time of amortized.  This gap may well shorten when working with “full” datasets instead of minibatching, as in cognitive modeling rather than deep learning tasks.
>
> > Can the authors briefly summarize the 'new' features of their algorithm relative to other PC ideas?
>
> Relative to contemporaries Langevin Predictive Coding and Monte Carlo Predictive Coding, to which we compare, we build a Gibbs sampling method out of the interpretation of predictive coding as gradient-based sampling. Relative to Pinchetti 2022, the first to generalize PC beyond Gaussian distributions and the Laplace assumption, we optimize a globally valid free energy/evidence lower bound, rather than a layer-wise one.  Relative to all previous PC methods, we optimize a tighter bound on the model evidence (by using Monte Carlo sampling instead of applying Jensen’s Inequality a second time under the Laplace assumption) and support approximate posterior distributions with no closed form at all, well beyond Gaussian approximate posteriors.
>
> > Why particle-based gradient descent as opposed to some other approach to empirical Bayes?
>
> Our motivation for choosing a gradient-based sampling method comes from our analogy between score functions in probability and prediction errors in neuroscience. We chose PGD in specific because it was a recent gradient-based particle method, and we chose particle methods out of a combination of neuroscientific and behavioral motivations: brains have been observed to solve non-conjugate Bayesian inference tasks. Also, compared to plain variational inference, both mean-field and amortized, particle methods achieve better likelihoods.  Tentatively, as one of our VAE baselines runs, it appears to be converging to a log-likelihood two orders of magnitude (10^2) away from what PPC achieves on the same experiment. We can give similar gaps for unpublished experiments.
>
> > Concretely, how might neurons learn log-density gradients?
>
> We have included the below text in our revised manuscript:
> Biological neurons often spike to represent changes in their membrane voltage, and some have even been tested and found to signal the temporal derivative of the logarithm of an underlying signal. Theorists have also proposed models under which single neurons could calculate gradients internally. In short, if neuronal circuits can represent probability densities, as many theoretical proposals and experiments suggest they can, then they can likely also calculate the prediction errors used in DCPC.
>
> > Is PPC strictly better than other PC algorithms, or is there some kind of tradeoff?
>
> There are two trade-offs: classical PC algorithms, being deterministic, can run more quickly than PPC/DCPC, and yet for the same reason, PPC/DCPC gives a far more expressive approximation to the true posterior distribution. This is the bullet we bite using particle algorithms and empirical distributions/particle clouds to represent the posterior.

---

> > ### Comment · Reviewer_AbQA · 2024-08-11
> >
> > I thank the authors for their helpful responses, and think they have mostly addressed my concerns. Overall, I think they make a worthy theoretical contribution to how the brain might implement inference over graphical models. I have increased my score.
> >
> > As a minor point, I encourage the authors to look over the paper for small typos (e.g., "hypotesis", line 25).

---

> > > ### Author Response · Authors · 2024-08-11
> > >
> > > We thank the reviewer for their feedback and the significant improvements it has aided in our manuscript.  We are finely combing the paper for similar small typos.

---

### Official Review · Reviewer_jNdT · 2024-07-16

**Soundness:** 4
**Presentation:** 3
**Contribution:** 3
**Rating:** 7
**Confidence:** 3

**Summary:**

The paper introduces a novel algorithm called Population Predictive Coding (PPC)
for structured generative models. The PPC algorithm aims to enhance the
performance and biological plausibility of predictive coding
approaches by respecting the correlation structure of generative models. The
paper provides theoretical foundations, discusses the biological plausibility of
PPC and performs empirical validation against previous predictive coding and
deep generative modeling algorithms.

**Strengths:**

### Originality

The proposed PPC algorithm is a novel combination of recent developments in
Monte Carlo sampling and the predictive coding approach. The transfer of these
new techniques to PC to derive the PPC algorithm is a substantial contribution.

### Quality

The technical derivation of the proposed method appears sound, but see my
disclaimer below. The empirical evaluation is thorough and the results are
convincing. Limitations, especially the higher computational costs are discussed
openly, which is great.

### Clarity

The paper is generally well-written and structured, making the high-level
concepts accessible to readers with a background in computational neuroscience
and machine learning. Especially the introduction and motivation are clear and
easy to follow.

The technical sections (2 and 3) are complex and could benefit
from additional diagrams, examples, and intuitive explanations. More detailed
breakdowns of the PPC algorithm steps would enhance comprehension, especially
for readers less familiar with predictive coding.

### Significance

The results are important and have the potential to significantly impact the
field of predictive coding and structured Bayesian inference. The proposed
algorithm addresses a difficult task and improves upon previous work, advancing
the state of the art demonstrably. The PPC algorithm's biological plausibility
has implications for both machine learning models and our understanding of
neural information processing.

Disclaimer: As an expert in simulation-based inference and with a background in
computational neuroscience, I found the high-level concepts and motivations of
the paper clear and compelling. However, my expertise in predictive coding is
limited, and I found the technical details in sections 2 and 3 challenging to
fully understand. This review reflects my understanding based on the provided
explanations and my background knowledge.

**Weaknesses:**

See text box above.

**Questions:**

1. I might be missing something here, but why is the FID score for PPC missing
   in table 3? On that note, have you considered using `cleanfid` instead of
   `pytorch-fid` to obtain more stable FID score estimates?
2. It is great that you implemented it all in the `Pyro` framework. However, the
   submission seems to be lacking an (anonymous) code attachment. Will you make
   the code publicly available with instructions and all relevant information
   for reproducing your experiments?

**Limitations:**

Yes, the authors have discussed the limitations of their work and potential negative societal impact.

---

> ### Author Rebuttal · Authors · 2024-08-07
>
> We thank the reviewer for the time and valuable feedback.
>
> > The technical sections (2 and 3) are complex.
>
> Thank you for the pointer. We have heavily modified the explanations, restructuring the sections and adding sentences that lead to a more intuitive understanding of the algorithm. For more details, we refer to the general answer provided above, and to the updated manuscript present in the anonymous link.
>
> > Have you considered using cleanfid instead of pytorch-fid to obtain more stable FID score estimates?
>
> We will add cleanfid to our calculations alongside our existing FID evaluator/implementation, for which we have been using torchmetrics by Lightning.ai.
>
> > It is great that you implemented it all in the Pyro framework. However, the submission seems to be lacking an (anonymous) code attachment. Will you make the code publicly available with instructions and all relevant information for reproducing your experiments?
>
> Yes. We have [anonymized our Github repository](https://anonymous.4open.science/r/ppc_experiments-8AF5/README.md); please share and enjoy. Our training code uses the Lightning framework and our evaluations consist of running a Jupyter notebook all the way through.  We will make the code publicly available and include a README.md specifically listing the combination of shell-command (for training) and notebook (for evaluation) to produce each and every figure and numerical result in the paper.

---

### Author Rebuttal · Authors · 2024-08-07

We sincerely thank the four reviewers for their close and detailed engagement with our manuscript, stretching across the computational neuroscience material and the core contributions on Bayesian inference.

We see the reviewers divided on score but in broad agreement on the contributions, and needs for improvement, in the paper. Overall, the reviewers agree on the technical soundness of the PPC algorithm, with R1 (jNdT) recognizing the connections to recent work in Sequential Monte Carlo, R2 (AbQA) appreciating the application of particle gradient descent, R3 (DHBv) seeing links to variational EM, and R4 (Cx3n) recognizing that Bayesian inference can apply to both discriminative and generative tasks. The reviewers also share our view that biological plausibility is a desirable goal to achieve in an inference or training algorithm.  Finally, they would all like to see the core technical sections of the paper significantly clarified. For all the reviewers, we now have an [anonymized Github repository](https://anonymous.4open.science/r/ppc_experiments-8AF5/README.md). To address some concerns, especially the ones regarding clarity and typos, we have decided to append a [new version of the manuscript](https://anonymous.4open.science/r/ppc_experiments-8AF5/neurips2024_population_predictive_coding.pdf) in the anonymous link. We will describe the changes below, as well as addressing the common concerns. We will start with the easiest concern to answer and then move on to the less trivial matters.

> Name PPC clashes with an existing acronym

We thank the reviewers for the reminder to pick a name that does not clash with “probabilistic population codes/coding”, particularly since both “old” PPC and “our” PPC concern Bayesian inference in the cortex.  We have taken the suggestion to simply change the name, instead choosing “Divide & Conquer Predictive Coding” to emphasize the connections with Gibbs sampling and [D&C-SMC](https://projecteuclid.org/journals/annals-of-applied-probability/volume-34/issue-1B/The-divide-and-conquer-sequential-Monte-Carlo-algorithm--Theoretical/10.1214/23-AAP1996.short). We have updated our manuscript accordingly in all usages of the name and acronym.

> Numerical and Figure Results on CelebA

We thank the reviewers for pointing out the broken figure links and typos in our initial manuscript submission.  The supplementary page to this response provides a fixed results figure for the generator network on CelebA experiment at 64x64 resolution, showing reconstructions on the left and de novo generations/samples on the right. It also includes a consolidated table of FID scores, showing that in an apples-to-apples comparison with particle gradient descent, PPC achieves a clear and significant improvement in FID. The PPC FID score remains worse as against Langevin Predictive Coding, because we have only recently managed to clarify with the authors their neural architecture, momentum-based optimization method, and likelihood function. Their [“discretised Gaussian” likelihood](https://github.com/lucidrains/denoising-diffusion-pytorch/blob/ec0a1c7596f654d9b5d3952a63ce3301128f1979/denoising_diffusion_pytorch/learned_gaussian_diffusion.py#L43) has been noted to produce sharper images and better FID scores than the continuous Gaussian log-density that we employed in training.  Running an apples-to-apples experiment will require implementing the “discretised Gaussian” as a Pyro distribution and thus take time into the reviewer discussion period, though we will of course do so. Finally, the neural architecture used for LPC here turns out to have come from the original Beta-VAE with slight modifications (nonlinearities and the likelihood function), and so we can train a Beta-VAE on the same problem to disentangle the training algorithm from the underlying neural architecture.

> Clarifying the technical presentation and novelty

R1 would like to see additional diagrams, examples, and intuitive explanations in Sections 2 and 3 of the paper. R2 appreciates the difficult task facing us in summarizing work on empirical Bayes, predictive coding and our novel contributions in these core sections, but would find it helpful if the sections more obviously separated between old and new contributions. R3 would like clarification on which aspects of the DCPC/PPC method are novel to our work specifically, rather than applications of the previous literature. We have endeavored to clarify Section 2 by separating it into clear paragraph-marked subsections for “Empirical Bayes”, “Predictive Coding”, and “Particle Algorithms”. The middle subsection includes an informal, but firm, definition of “predictive coding” taken from Salvatori et al 2023, which allows us to compare and contrast DCPC/PPC with that definition, thus with the existing literature, in Section 3. We have also endeavored to clarify Section 3 and the construction of the DCPC algorithm in several ways, including discussing how DCPC fits, in some ways, and extends, in others, the definition of predictive coding from Section 2.

> Clarifying the novelty of the algorithm vs microcircuit hypothesis

The reviewers point out that it is difficult from the submitted manuscript to determine which aspects of the PPC method are new, versus coming from published work, and thus whether the PPC paper’s core contribution is the algorithm itself or its mapping onto neuronal circuits. We end our clarifications of Section 2 by discussing the limitations in applying existing particle algorithms, samplers, and variational methods in targeting the joint density of a large-scale, structured graphical model, the better to motivate our novel structured PC algorithm. In the algorithmic details, Kuntz 2023 did not construct or evaluate importance weights, while Lindsten 2017/Kuntz 2024 and Naesseth 2015 did not propose a unique decomposition of a target model into Gibbs kernels as target densities for samplers, nor did they employ gradient-based proposals.

---

> ### Author Response · Authors · 2024-08-13
>
> Dear reviewers,
>
> Thank you again for your time and comments, and for having been amenable to discussion during the rebuttal period. This message is to communicate a nice improvement in our experimental evaluation: in the last days, we have also run an apples-to-apples comparison against the Langevin PC work [1], using the same model, and, more importantly, the discretized Gaussian function as a likelihood function, that they do. The results show that the discretized Gaussian largely improves the performance in our case as well.
> In terms of results, the authors report three different numbers for three different methods: prior sampling, and two amortized warm-start techniques, the best one using Jeffrey’s Divergence. However, we decided to not implement warm-start techniques, as they are not biologically plausible in the sense that we mean in our work, and hence only perform prior sampling. Despite that, D&C-PC achieves a FID score of 96.0473 (+/- 0.2667), that is (1) much better than the one of 120 they report for prior sampling, in the apples-to-apples comparison; and (2) better than both of the method that use warm starts, where they report a FID score on the test set of 97.49 (Table 1 in [1]).
>
> We can thus claim that D&C PC is the variation of PC that achieves the best results in the literature when it comes to image generation tasks, even when compared against less bio-plausible PC methods. We have updated the manuscript accordingly.
> A second, minor, change that we have applied to the manuscript, is to write the free energy in terms of a product of complete conditionals as the recognition model Q and then consider estimating it by Monte Carlo with the whole Langevin to nested SMC procedure, as suggested by reviewer DHBv. We agree that this would further improve the clarity of our work.
>
> We once again deeply thank the reviewers for their close engagement that has done so much to improve the paper.
>
> [1] Zahid, U., Guo, Q. and Fountas, Z., Sample as you Infer: Predictive Coding with Langevin Dynamics. In Forty-first International Conference on Machine Learning, ICML 2024.

---

### Decision · Program_Chairs · 2024-09-25

**Decision:**

Accept (poster)

**Comment:**

The paper presents a novel algorithm, Population Predictive Coding (PPC), which aims to enhance both the performance and biological plausibility of predictive coding by incorporating  Monte Carlo sampling techniques. The reviewers generally agree that the paper makes a meaningful contribution to the field, offering a fresh approach that integrates ideas from machine learning and neuroscience to address the limitations of existing predictive coding methods. While the reviewers raise concerns about the complexity of some explanations, missing results, and the need for clearer documentation and presentation, the authors have effectively responded to these critiques by clarifying their contributions, updating the manuscript (particularly regarding biological plausibility), and providing additional experimental results. Although the overall connection with neural circuitry is intriguing, the paper would benefit from a more direct comparison with neural data. Nevertheless, given the technical soundness, moderate-to-high impact, and the authors' responsiveness to feedback, the paper represents a valuable advancement in understanding predictive coding and structured Bayesian inference.